# Coordinating meiotic prophase I progression and early oocyte differentiation

Kimberly M. Abt[1], Myles A. Bartholomew[1], Anna E. K. Nixon[1], Hanna E. Richman[2], Megan A. Gura[1], Kimberly A. Seymour[2] and Richard N. Freiman[1,2,*]

## ABSTRACT

Female reproductive senescence results from the regulated depletion of a finite pool of oocytes called the ovarian reserve. This pool of oocytes is initially established during fetal development, but the oocytes that it consists of must remain quiescent for decades until they are activated during maturation in adulthood. In order for developmentally competent oocytes to populate the ovarian reserve, they must successfully initiate both meiosis and oogenesis. As the factors that regulate the timing and fidelity of these early events remain elusive, we assessed the precise function and timing of the transcriptional regulator TAF4b during meiotic prophase I progression in mouse fetal oocytes. Compared to matched controls, E14.5 *Taf4b*-deficient oocytes enter meiosis I in a timely manner; however, their subsequent progression through the pachytene-to-diplotene transition of meiotic prophase I is compromised. Moreover, this disruption of meiotic progression is associated with the reduced ability of *Taf4b*-deficient oocytes to repair double-strand DNA breaks. Transcriptional profiling of *Taf4b*-deficient oocytes reveals that between E16.5 and E18.5 these oocytes fail to properly coordinate the reduction of meiotic gene expression and the activation of oocyte differentiation genes.

KEY WORDS: Meiosis, Oogenesis, Transcription, Ovarian reserve, Fetal oocyte attrition, Female infertility

## INTRODUCTION

The formation of a functional ovarian reserve during embryonic development is essential to female fertility in adulthood (Grive and Freiman, 2015). This finite pool of gametes will be continually depleted throughout the reproductive lifespan of females until the onset of menopause, which occurs, on average, at 50±4 years of age. Onset of menopause before 40 years of age is considered premature, and this leads to infertility as well as an increased risk of mortality (van Noord et al., 1997). Premature menopause is clinically defined as primary ovarian insufficiency (POI), but the cause of infertility cannot be determined in most cases. Individuals with POI display a diminished ovarian reserve measured by reduced serum anti-Mullerian hormone (AMH) levels, but the timing and mechanisms of this fertility deficit remain unknown (Cordts et al., 2011; Cox and

Liu, 2014; Nelson, 2009; Ruth et al., 2021). POI also disrupts key endocrine functions of the ovary, and this has detrimental effects on the health of extra-ovarian organs, including the bones, heart and brain (Rossetti et al., 2017). Defective ovarian reserve establishment is a likely contributor to the onset of POI, but the molecular mechanisms that govern this fundamental process are still being uncovered.

The initiation of meiosis and early oogenesis are two of the most crucial processes that must occur during the establishment of the ovarian reserve (Grive and Freiman, 2015). Oocytes are first formed during fetal development in cysts that break down to produce individual primordial follicles that consist of a single dictyate-arrested oocyte surrounded by a flattened layer of pre-granulosa cells (Pepling, 2012). These primary oocytes can be arrested for decades until after puberty, when ovulation triggers the completion of meiosis I and fertilization promotes the completion of meiosis II (Bolcun-Filas and Handel, 2018; Cohen and Holloway, 2015). The major developmental events of early oogenesis, such as cyst breakdown and fetal oocyte meiosis I arrest, are conserved between mice and humans. Thus, dissecting conserved genetic pathways underlying the embryonic development of the initial ovarian reserve between the mouse and human ovary is crucial to developing better tools to detect a reduction of the ovarian reserve. These insights are key for devising novel strategies to preserve declining fertility and endocrine functions in individuals with POI.

TATA-box binding protein associated factor 4b (TAF4b) was originally identified as a gonadal-enriched subunit of the TFIID transcription initiation complex and a paralog of TAF4a in the mouse genome (Freiman et al., 2001). Adult female *Taf4b*-deficient mice model crucial aspects of POI in women, such as elevated serum levels of follicle stimulating hormome (Lovasco et al., 2010). TAF4b is crucial for development of a healthy ovarian reserve in the mouse embryonic ovary, and *Taf4b*-deficient mice have a reduced number of primordial follicles immediately after birth (Grive et al., 2014). Furthermore, genes and pathways whose expression depends on TAF4b in oocytes significantly overlap those disrupted in Turner syndrome and Fragile X-associated POI (FX-POI), two prominent genetic examples of POI (Gura et al., 2022). In the mouse, we determined that *Taf4b* mRNA and protein are consistently enriched in the germ cells of the embryonic ovary compared to the somatic cells from embryonic day (E)11.5 to E18.5 (Gura et al., 2020). We further demonstrated that *Taf4b* mRNA expression significantly increases during meiotic differentiation, as do other alternative TFIID genes, including *Taf7l* and *Taf9b* (Gura et al., 2020). Most recently, we identified 570 TAF4b occupancy peaks using CUT&RUN in E16.5 mouse oocytes, of which 94% were located in the core promoter region just upstream of the transcription start site (TSS). Importantly, TAF4b peaks were identified in a large number of crucial meiotic, oogenesis and POI-related gene promoters, including *Fmr1* and *Dazl* (Gura et al., 2022). It is still unknown if and how meiotic events and oocyte differentiation are coordinated to establish a healthy ovarian reserve.

[1]MCB Graduate Program, Brown University, 70 Ship Street, Box G-E4, Providence, RI 02903, USA. [2]Department of Molecular Biology, Cell Biology, and Biochemistry, Brown University, 70 Ship Street, Box G-E4, Providence, RI 02903, USA.

*Author for correspondence (richard_freiman@brown.edu)

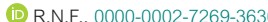 R.N.F., 0000-0002-7269-3630

**DEVELOPMENT**

A seminal study by Dokshin et al. in 2013 concluded that meiosis and oocyte growth and differentiation are genetically separable events in the mouse (Dokshin et al., 2013). As the progression of prophase I and oocyte differentiation overlap, it would be intriguing to find a common regulator of both processes. Meiotic prophase I is a complex process that occurs semi-synchronously in the mouse ovary from E14.5 to postnatal day (P) 3 and can be broken down into five substages (leptonema, zygonema, pachynema, diplonema and dictyate) based on the cytological appearance of the oocyte chromosomes (Cohen et al., 2015). Here, we show that both *Taf4b* mRNA and protein are enriched during the pachytene stage of meiotic prophase I in fetal oocytes. While *Taf4b*-deficient oocytes are able to initiate meiosis I in a timely fashion, their subsequent progression is delayed and they largely fail to progress from pachynema to diplonema. Meiotic progression delays in the absence of TAF4b are coincident with increased RAD51 and γH2AX DNA damage foci, suggesting that elevated levels of DNA damage underlie the excessive loss of these oocytes shortly after birth. Surprisingly, we observed that most *Taf4b*-deficient oocytes reached pachynema at E18.5, allowing us to compare gene expression of FACS-sorted control versus *Taf4b*-deficient pachynema-enriched oocytes. At this timepoint, we discovered that *Taf4b*-deficient oocytes display increased expression of meiotic genes and decreased expression of oocyte differentiation genes compared to controls. Together, these data suggest that TAF4b may coordinate the timely reduction of meiotic genes and upregulation of oogenesis genes required to establish a robust ovarian reserve. Although these two pathways are genetically separable, it is of great interest if a single transcriptional regulatory factor, TAF4b, simultaneously integrates both crucial functions.

## RESULTS

### TAF4b expression is enriched during pachynema in female germ cells

*Taf4b* mRNA is enriched in female germ cells shortly after meiotic initiation, which led us to examine whether TAF4b is enriched at a particular substage of meiotic prophase I. To accomplish this, we integrated multiple publicly available single cell RNA-sequencing (scRNA-seq) datasets that spanned ovarian reserve establishment in the mouse from meiotic initiation to primordial follicle formation. To perform the dataset integration, we reprocessed four datasets listed in Fig. 1A, three of which performed scRNA-seq on whole ovaries and one of which performed scRNA-seq on Oct4-GFP⁺ oocytes (Ge et al., 2021; Niu and Spradling, 2022; Wang et al., 2020a; Zhao et al., 2020). We first generated a merged Seurat object of all cells from each dataset. We then selected for high-quality cells based on parameters defined in Fig. S1A,B. We used Monocle3 to perform initial dimensionality reduction, and found that the majority of Oct4-GFP⁺ oocytes from GSE130212 clustered with *Dazl*- and *Ddx4*-positive germ cells obtained from the three scRNA-seq datasets on whole ovaries (Fig. S1C,E,F). Lastly, germ cells isolated at similar timepoints from different datasets clustered together, suggesting the integration was successful (Fig. S1D).

Given the majority of *Ddx4*- and *Dazl*-positive cells clustered together in partition 1 (Fig. S1G), this subset of cells was used for our downstream pseudotime analysis (Fig. 1B). We found that our pseudotime analysis successfully recapitulated gene expression changes that occur throughout prophase I and that chronological time points align well with pseudotime in the UMAP (Fig. 1B,C). For example, expression of *Stra8*, which is a master regulator of meiotic initiation, peaks at the beginning of the pseudotime course (Anderson et al., 2008). *Rec8*, a cohesin protein that supports DNA replication prior to meiosis, peaks at a similar point along the

pseudotime trajectory (Dokshin et al., 2013). Expression of genes required for recombination (*Msh4* and *Msh5*) and crossover formation (*Mlh1* and *Mlh3*) peak after meiotic initiation (Bolcun-Filas and Handel, 2018). Furthermore, expression of oocyte-enriched genes that are required for primordial follicle survival, such as *Sohlh1* and *Uchl1*, peak at the end of the pseudotime trajectory (Niu and Spradling, 2022; Pangas et al., 2006; Woodman et al., 2022). We then evaluated the pseudotime trajectory of two germ cell-enriched components of the TFIID complex: *Taf4b* and *Taf7l* (Gura et al., 2020). Expression of both increased in the section of the pseudotime trajectory enriched for zygotene and pachytene markers, including *Msh4*, *Msh5*, *Mlh1* and *Mlh3*, consistent with these stages of meiotic prophase I (Fig. 1C,D).

To determine whether TAF4b protein is enriched in pachynema, we performed quantitative immunofluorescence on prophase I chromatin spreads isolated from E16.5 and E18.5 ovaries. Preparing chromatin spreads from fetal ovarian tissue allows us to both identify meiotic germ cells and characterize their substage based on the configuration of synaptonemal complex protein 3 (SYCP3) (Grive et al., 2016; Hwang et al., 2018). By staining chromatin spreads for TAF4b and DAPI, in addition to SYCP3, we were able to use a recently published method to quantify the amount of TAF4b present relative to the DNA content of the cell in oocytes at specific stages of prophase I (Alexander et al., 2023). In E16.5 wild-type mice, we found oocytes in leptonema, zygonema or pachynema, and we observed that the staining pattern of TAF4b is diffuse throughout the chromatin spreads at each stage (Fig. 2A). To validate our TAF4b antibody is specific in this assay, we performed immunofluorescence on prophase I spreads isolated from wild-type (*Taf4b*⁺/⁺) and *Taf4b*-deficient (*Taf4b*⁻/⁻) mice at E18.5, and we detected high levels of TAF4b signal in wild-type oocytes, but not in our *Taf4b*⁻/⁻ spreads (Fig. S2). By measuring fluorescence intensity of TAF4b in E16.5 chromatin spreads from wild-type mice, we found that its expression is significantly higher in pachytene oocytes even though it is present at all stages observed, which agrees with our findings from the scRNA-seq integration (Fig. 2B). We performed a similar analysis in wild-type oocytes at E18.5, and we found that TAF4b expression is significantly higher in pachytene versus diplotene oocytes (Fig. 2C,D). Together, these data suggest that expression of *Taf4b* mRNA and protein are highest in pachytene oocytes during ovarian reserve establishment.

### *Taf4b* deficiency does not impede meiotic initiation at E14.5

Although Taf4b expression peaks during pachynema, both *Taf4b* mRNA and protein are detectable during earlier stages of meiotic prophase I, raising the possibility that *Taf4b* may function during meiotic initiation. To test whether entry into meiosis is perturbed in the absence of *Taf4b*, we examined fetal ovaries at E14.5, a developmental time point corresponding to initiation of meiotic S phase in mouse oocytes.

Ovarian tissue sections from E14.5 *Taf4b*⁺/⁺ and *Taf4b*⁻/⁻ embryos were immunostained for germ cell nuclear antigen (GCNA) to mark germ cells and STRA8, as a marker of meiotic entry. We quantified the density of GCNA-positive germ cells and STRA8-positive cells by normalizing cell counts to the DAPI-positive ovarian area. At E14.5, *Taf4b*⁺/⁺ and *Taf4b*⁻/⁻ ovaries were indistinguishable, with no significant differences in total germ cell density or in the proportion of STRA8-positive cells between genotypes (Fig. 3A-D). These findings are consistent with our previous bulk RNA-seq analysis of E14.5 Oct4-GFP⁺ oocytes isolated from *Taf4b*⁺/⁺ and *Taf4b*⁻/⁻ ovaries, which found little to no differential gene expression between the two genotypes at this timepoint (Gura et al., 2022). Together, these histological and transcriptomic analyses demonstrate that

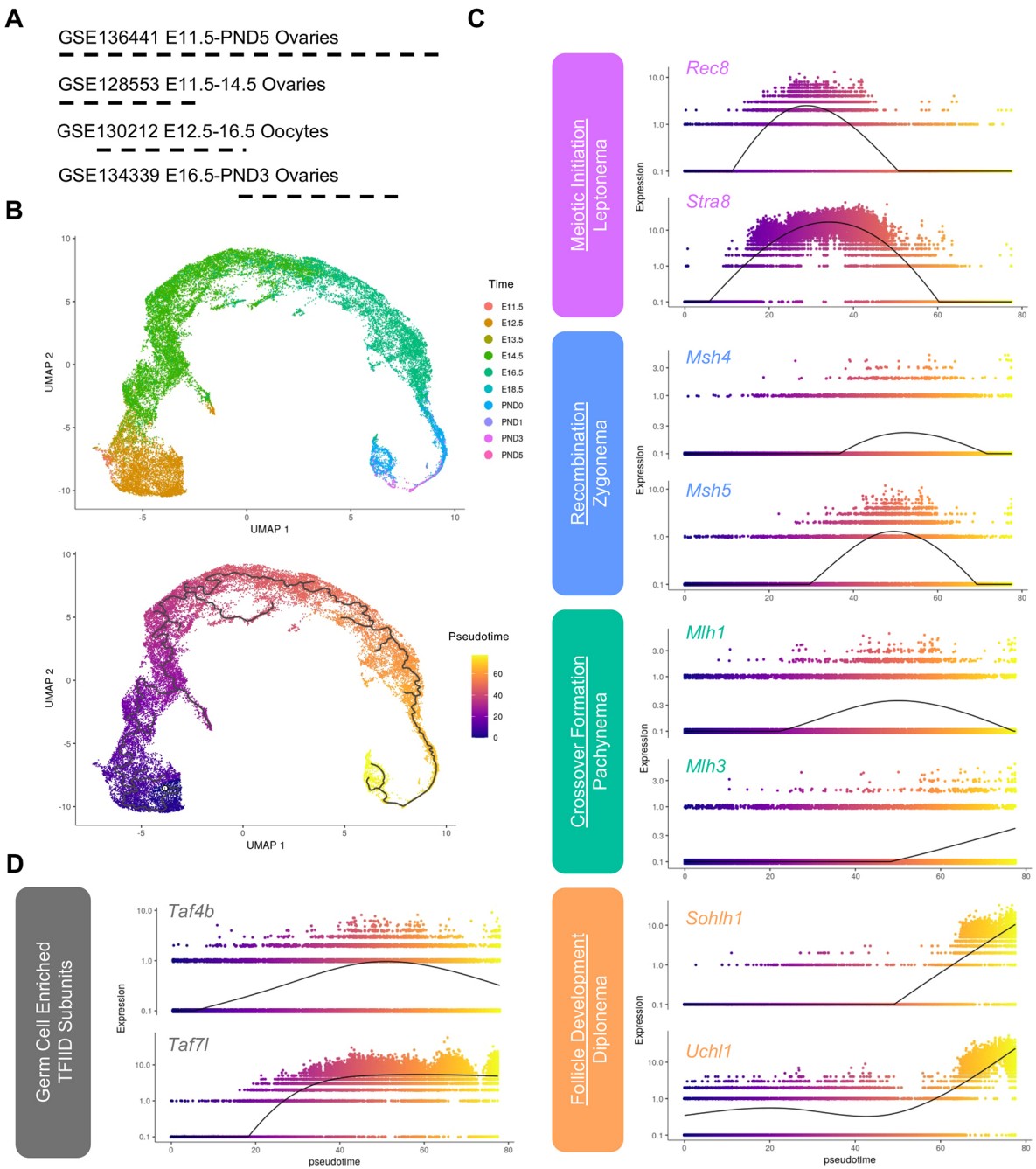

**Fig. 1. Integration of scRNA-seq data from oocytes during ovarian reserve establishment.** (A) Schematic of datasets used for integration. (B) Uniform manifold approximation and projection (UMAP) of female germ cells colored by time (top panel) and pseudotime (bottom panel). (C) Expression of candidate genes plotted in terms of pseudotime. Colors of gene labels indicate the approximate stage of prophase I. (D) Expression of germ cell-enriched TFIID components plotted in terms of pseudotime.

meiotic initiation, defined here as entry into the meiotic program prior to leptotene, occurs normally in the absence of *Taf4b*.

## *Taf4b*-deficient oocytes experience delayed meiotic prophase I progression

While E14.5 germ cells can initiate meiosis properly in the absence of *Taf4b*, we questioned whether there was a delay in prophase I progression in TAF4b-deficient embryonic oocytes. To further examine the precise timing of prophase I progression in *Taf4b*$^{-/-}$ oocytes prior to and immediately after birth, we prepared chromatin spreads from three *Taf4b*$^{+/+}$ and *Taf4b*$^{-/-}$ mice at E16.5 and E18.5,

as well as at P0 and stained for SYCP3 to determine the proportion of oocytes in each substage of prophase I. We isolated 50-100 spreads per animal and then performed a Chi-squared test to determine whether there was a significant difference in the percentage of oocytes at each stage between the two genotypes. At E16.5, *Taf4b*$^{-/-}$ spreads oocytes do not reach pachytene as efficiently as wild-type controls (Fig. 4A). Intriguingly, the majority of *Taf4b*$^{-/-}$ oocytes reach pachynema by E18.5, but ∼20% of *Taf4b*$^{-/-}$ germ cells remain in zygonema, whereas a similar percentage of *Taf4b*$^{+/+}$ oocytes have proceeded to diplonema (Fig. 4A). By P0, most *Taf4b*$^{+/+}$ oocytes are in diplonema, but 80% of *Taf4b*$^{-/-}$ germ cells remain in pachynema

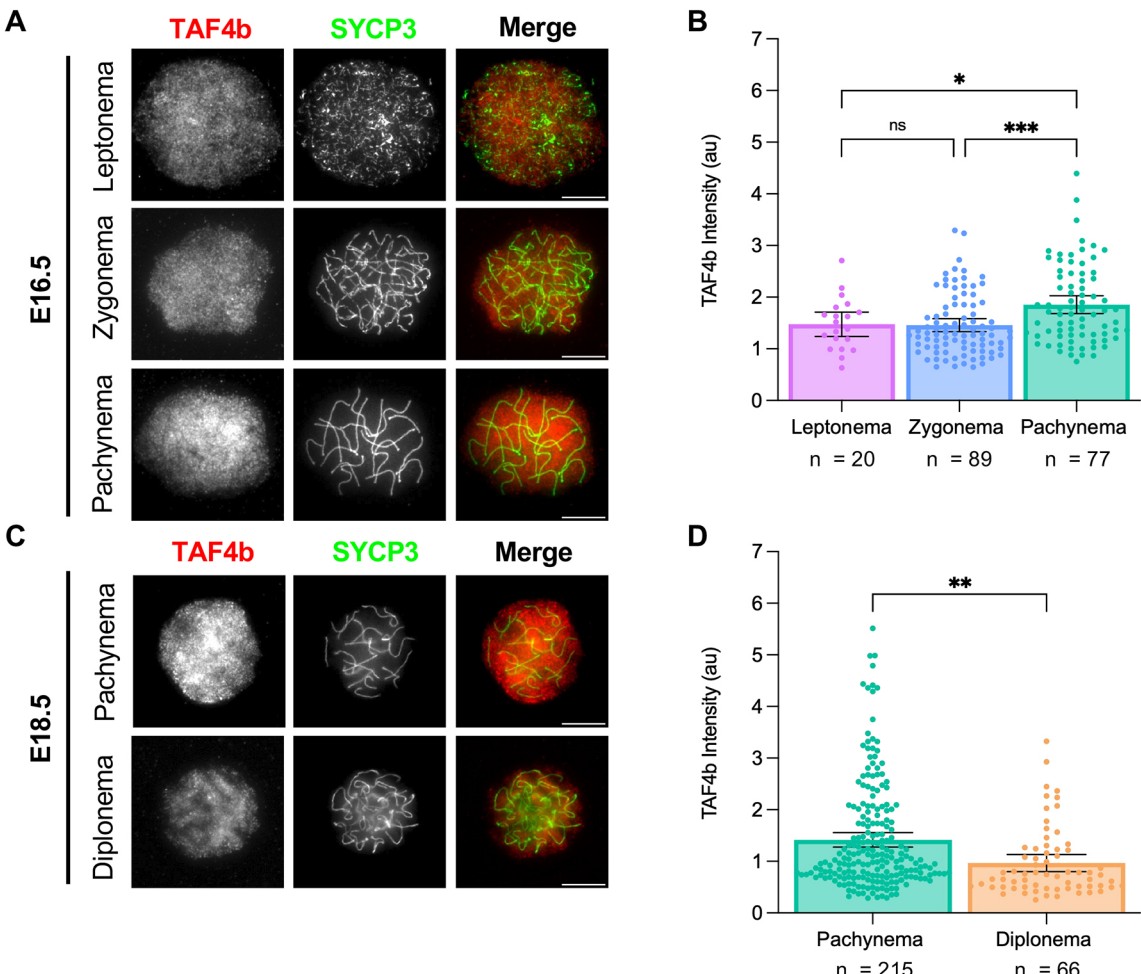

**Fig. 2. TAF4b is enriched in pachytene oocytes at E16.5 and E18.5.** (A,C) Prophase I chromatin spreads were prepared from wild-type ovaries at E16.5 and E18.5 using the drying down technique. Images of oocytes stained with TAF4b (red) and SYCP3 (green) at each prophase I stage present at E16.5 (A) or E18.5 (C). Scale bars: 10 μm. (B,D) Quantification of TAF4b signal intensity in oocytes grouped by stage at E16.5 (B) or E18.5 (D). au, arbitrary units. Spreads were collected from three animals at E16.5 and four animals at E18.5. *n* refers to the number of spreads used in analysis. Data are mean±s.e.m. Dots represent individual spread results. Statistical significance for stage-specific comparison at E16.5 (C) was determined using an ordinary one-way ANOVA with multiple comparisons. For E18.5 samples (D), an unpaired two-tailed *t*-test was used. ns, not significant; *$P<0.05$, **$P<0.01$, ***$P<0.001$.

(Fig. 4A). Together, these data indicate that *Taf4b*$^{-/-}$ oocytes are unable to progress through prophase I in a timely manner and are largely stalled in pachynema at P0.

### *Taf4b*$^{-/-}$ oocytes have persistent DNA damage during pachynema and diplonema

During fetal development and early postnatal life, there is a significant amount of fetal oocyte attrition (FOA) in the mammalian ovary, likely driven by quality control processes that eliminate defective germ cells and consequently ensure the survival of the highest quality oocytes (Grive, 2020; Grive and Freiman, 2015; Hunter, 2017). Errors in prophase I, such as persistent DNA damage and chromosomal asynapsis, are known triggers of excessive FOA (Di Giacomo et al., 2005; Huang and Roig, 2023; Ravindranathan et al., 2022; Rinaldi et al., 2017). We have previously shown that *Taf4b*$^{-/-}$ oocytes have increased levels of asynapsis at E16.5, but it is unknown whether there is excessive DNA damage in the absence of TAF4b (Grive et al., 2016; Gura et al., 2022). To answer this question, we stained prophase I spreads from *Taf4b*$^{+/+}$ and *Taf4b*$^{-/-}$ with antibody against γH2AX as a marker of unrepaired double strand breaks (DSBs). We pooled spreads from three timepoints (E16.5, E18.5 and P0) and separated spreads by substage

in order to compare levels of γH2AX between *Taf4b*$^{+/+}$ and *Taf4b*$^{-/-}$ oocytes using quantitative immunofluorescence. We found that *Taf4b*$^{+/+}$ and *Taf4b*$^{-/-}$ oocytes had similar levels of γH2AX intensity during zygonema; however, *Taf4b*$^{-/-}$ oocytes have elevated levels of γH2AX intensity during pachynema and diplonema (Fig. 5A-F). We did not compare levels of γH2AX intensity between *Taf4b*$^{+/+}$ and *Taf4b*$^{-/-}$ oocytes in leptonema because there are not sufficient numbers of *Taf4b*$^{+/+}$ oocytes in this stage at the timepoints analyzed. We also quantified the number of RAD51 foci in *Taf4b*$^{+/+}$ and *Taf4b*$^{-/-}$ oocytes at E16.5 to evaluate single strand invasion during homologous recombination. We found a similar number of RAD51 foci in *Taf4b*$^{+/+}$ and *Taf4b*$^{-/-}$ oocytes during leptonema as well as zygonema (Fig. S3A-D). However, in *Taf4b*$^{-/-}$ pachytene oocytes, we observed an increased number of RAD51 foci compared to *Taf4b*$^{+/+}$ (Fig. S3E,F). These data suggest that levels of DNA damage are normal in *Taf4b*$^{-/-}$ oocytes during the initial stages of prophase I, but that they remain elevated and persistent during pachynema and diplonema, when wild-type oocytes are completing DNA repair. This is consistent with our findings that *Taf4b*$^{-/-}$ oocytes largely initiate prophase I similarly to controls, but are not able to progress to later stages in a timely manner.

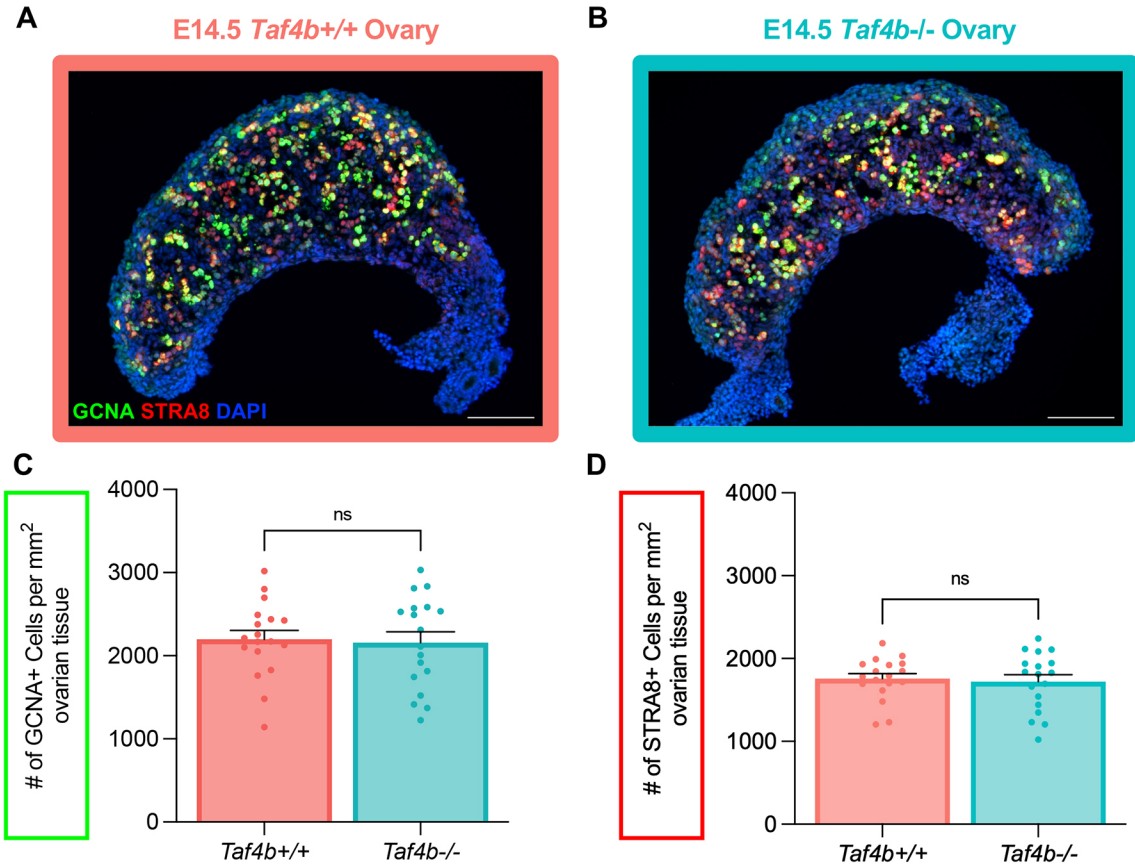

**Fig. 3. *Taf4b*<sup>−/−</sup> ovaries initiate meiosis normally at E14.5.** (A,B) Images of E14.5 *Taf4b*<sup>+/+</sup> (A) and *Taf4b*<sup>−/−</sup> (B) ovary sections stained with GCNA (green) as a marker of total germ cells, STRA8 (red) as a marker of meiotic germ cells and DAPI (blue) as a marker of cell nuclei. Scale bars: 100 µm. (C,D) Quantification of GCNA (C) or STRA8 (D) cell density of tissue sections per genotype. Cell density was calculated by dividing the number of positively marked cells by the area of the tissue section. Sections were collected from three E14.5 ovary pairs per genotype. Data are mean±s.e.m. Each dot represents the number of positive oocytes counted in a single ovarian section. Statistical significance for genotype comparisons was determined using an unpaired two-tailed *t*-test. ns, not significant.

## TAF4b promotes the pachytene-diplotene transition in E18.5 oocytes

As *Taf4b* mRNA expression is enriched in pachynema and *Taf4b*<sup>−/−</sup> oocytes remain in this stage for an extended period of time before they are culled, it is essential to understand what genes are regulated by TAF4b at this meiotic sub-stage. To accomplish this goal, we performed bulk RNA-seq on sorted Oct4-EGFP<sup>+</sup> oocytes from five *Taf4b*<sup>+/+</sup>, *Taf4b*<sup>+/−</sup> and *Taf4b*<sup>−/−</sup> ovary pairs at E18.5 (cell numbers for each RNA-seq sample can be found in Fig. S4). Although *Taf4b*<sup>−/−</sup> oocytes still display a lag in prophase I progression at E18.5, both *Taf4b*<sup>+/+</sup> and *Taf4b*<sup>−/−</sup> oocytes are largely in pachynema (Fig. 4). We included *Taf4b*<sup>+/−</sup> oocytes in addition to *Taf4b*<sup>+/+</sup> oocytes as controls because we found that *Taf4b*<sup>+/−</sup> fetal oocytes have a slight delay in prophase I progression, even though the adults are fertile (Freiman et al., 2001). The resulting principal component analysis (PCA) plot shows the *Taf4b*<sup>−/−</sup> samples clustering together away from both *Taf4b*<sup>+/+</sup> and *Taf4b*<sup>+/−</sup> control samples (Fig. 6A). We identified 1551 differentially expressed genes (DEGs) between *Taf4b*<sup>+/+</sup> or *Taf4b*<sup>+/−</sup> and *Taf4b*<sup>−/−</sup> oocytes, which were defined as protein-coding, average transcripts per million (TPM)>1, *P*adj<0.05 and log2FoldChange >l0.6 l (Fig. 6B, Table S1). From this list of DEGs, 1070 were decreased (downregulated) and the remaining 481 were increased (upregulated).

To determine which of these transcriptional changes reflect direct TAF4b-dependent regulation, we compared E18.5 DEGs with previously published TAF4b CUT&RUN peaks identified at E16.5

(Gura et al., 2022). This analysis revealed that a subset of 50 E18.5 DEGs overlap with E16.5 TAF4b CUT&RUN peaks (Fig. 6E). Notably, several meiotic genes crucial for recombination and prophase I progression, including *Meioc*, *Msh5* and *Mdc1*, exhibit reproducible TAF4b occupancy coincident with H3K4me3 enrichment at their regulatory regions, consistent with direct transcriptional regulation by TAF4b (Fig. S7). Because TAF4b chromatin occupancy is likely dynamic across meiotic substages, performing CUT&RUN at E18.5 to directly assess stage-matched binding will be an important future direction for research.

Together, these data suggest that, while a subset of E18.5 transcriptional changes coincide with TAF4b chromatin occupancy, loss of *Taf4b* is associated with broader alterations in gene expression programs at this stage. As elevated DNA damage and asynapsis results in transcriptional silencing, the higher number of downregulated genes is consistent with our finding of increased DNA damage in *Taf4b*<sup>−/−</sup> oocytes (Fig. 5 and Fig. S3) (Cloutier et al., 2015). Moreover, if TAF4b acts solely as a transcriptional activator, we would expect more genes to be downregulated in its absence. Strikingly, *Uchl1* and *Sohlh1* were two of the most significantly downregulated genes (Fig. 6B). We also found other well characterized oogenesis genes, such as *Nobox*, *Figla*, *Kit*, *Lhx8*, *Foxo3* and *Zp3* were downregulated in E18.5 *Taf4b*<sup>−/−</sup> oocytes (Choi et al., 2008; Jones and Pepling, 2013; Rajkovic et al., 2004; Shimamoto et al., 2019; Soyal et al., 2000; Suzumori et al.,

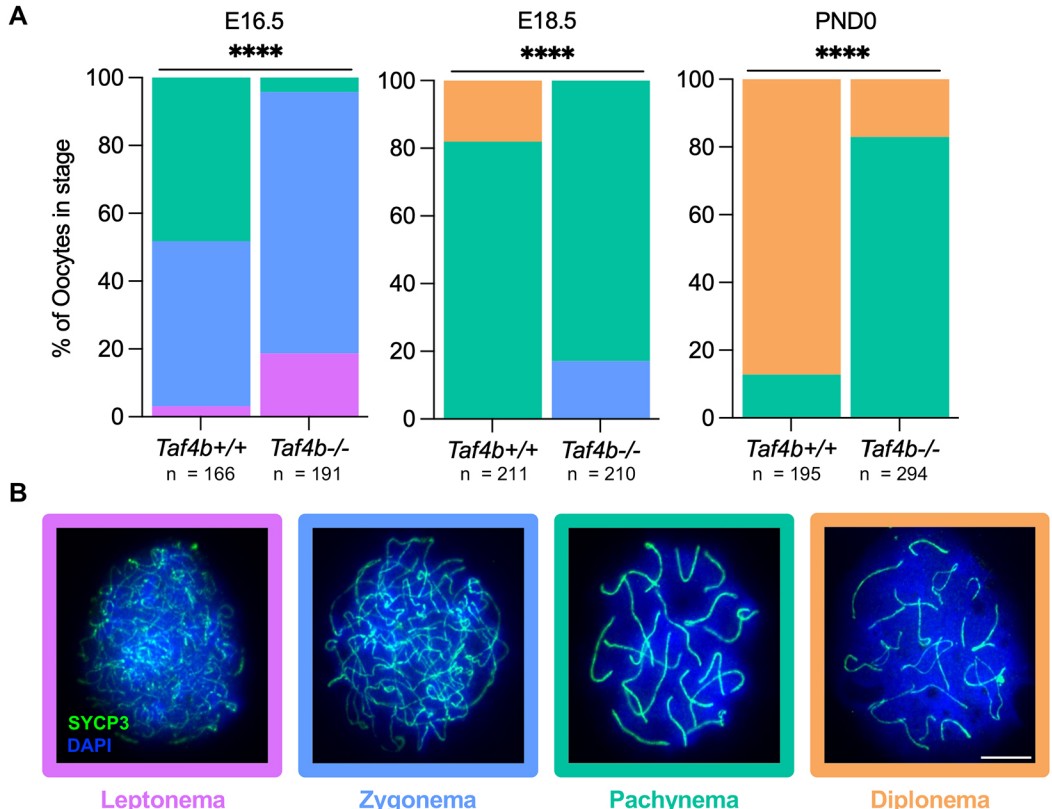

**Fig. 4. *Taf4b*<sup>−/−</sup> ovaries experience delays in prophase I progression at E16.5, E18.5 and postnatal day 0**. (A) The proportion of *Taf4b*<sup>+/+</sup> and *Taf4b*<sup>−/−</sup> chromatin spreads found at each substage of prophase I at E16.5, E18.5 and postnatal day 0. Substage was determined by spatial configuration of SYCP3 and representative images of spreads at each stage are shown in B. Spreads were stained with SYCP3 (green) and DAPI (blue). Scale bar: 10 µm. Colors of labels in the stacked bar chart correspond to prophase I stage: leptonema ( purple), zygonema (blue), pachynema (teal) and diplonema (orange). Spreads were collected from three animals per genotype at each timepoint. *n* refers to the number of spreads analyzed per stage. Representative images of zygonema and diplonema in B are also shown in Fig. 5B,F. Statistical significance was calculated using a Chi-square test; ****P<0.0001.

2002; Wang et al., 2020b). Finding *Nobox* and *Figla* as DEGs corroborates our previous research that TAF4b directly binds to the promoter regions of these genes in E18.5 wild-type ovaries (Grive et al., 2016). In addition, H2AX was one of the most significantly upregulated genes, and we have shown its phosphorylated protein product (γH2AX) is elevated in *Taf4b*<sup>−/−</sup> oocytes during pachynema and diplonema (Fig. 5).

To gain a comprehensive understanding of how gene expression is perturbed in E18.5 *Taf4b*<sup>−/−</sup> oocytes, we performed gene ontology (GO) analysis of all 1551 DEGs (Fig. 6C). Multiple GO categories associated with oogenesis and meiosis were enriched in our DEGs, allowing us to prepare heatmaps of genes lists associated with each process to examine how their expression was altered in *Taf4b*<sup>−/−</sup> oocytes (Fig. 7A,B). We found that genes involved in meiotic recombination (*Spo11*, *Dmc1*, *Meiob*, *Msh4*, *Brme1*, *Hormad1*, *M1ap* and *Mcmdc2*) and meiosis-specific telomere movements (*Spdya* and *Majin*) were upregulated in E18.5 *Taf4b*<sup>−/−</sup> oocytes compared to controls (Fig. 7A) (Bolcun-Filas and Handel, 2018). Expression of these genes could be elevated in *Taf4b*<sup>−/−</sup> oocytes because they may be in an earlier state of pachytene than controls. Interestingly, recent work has shown that *Meioc* is required for the extension of prophase I, and it is possible that upregulation of this gene at E18.5 in *Taf4b*<sup>−/−</sup> oocytes plays a role in the observed prophase I lag (Abby et al., 2016; Soh et al., 2017). We also found some genes involved in meiosis and DNA recombination were downregulated (*Mnd1*, *Mus81* and *Ercc1*) in the absence of *Taf4b*. Dysregulation of genes involved in DNA recombination is

particularly interesting given that pachytene *Taf4b*<sup>−/−</sup> oocytes have reduced numbers of MLH1 foci and fail to properly form crossovers at E18.5 (Grive et al., 2016). When looking at DEGs related to oogenesis and female gamete generation, we noticed that most of the upregulated genes were also associated with meiosis (*Hormad1*, *Spo11*, *Spdya*, *Meioc* and *Mcmdc2*), whereas genes associated with oocyte differentiation were downregulated (*Nobox*, *Figla*, *Sohlh1*, *Foxo3* and *Zp3*) (Fig. 7B). Additional factors important for early oocyte development, such as *Uchl1*, *Kit*, *Lhx8* and *Trp63* are similarly downregulated in *Taf4b*<sup>−/−</sup> oocytes (Fig. 6B, Table S1). Furthermore, we compared our E18.5 *Taf4b*<sup>−/−</sup> DEGs with a list of nearly 200 genes found to be enriched in diplotene/dictyate oocytes using scRNA-seq, and the majority of these genes were downregulated in the absence of Taf4b (Figs 6D and 7B). This is consistent with our findings that few *Taf4b*<sup>−/−</sup> oocytes are in diplonema at E18.5, and that TAF4b may play a role in regulating expression of genes required to reach diplonema. Overall, E18.5 *Taf4b*<sup>−/−</sup> oocytes fail to properly express correct levels of meiotic genes and are unable to activate expression of genes required for oocyte differentiation and growth.

To understand how gene expression is perturbed in the absence of TAF4b over time, we compared our bulk RNA-seq data from E16.5 (Gura et al., 2022) and E18.5 (this study) *Taf4b*<sup>−/−</sup> oocytes. We reprocessed our E16.5 *Taf4b*<sup>+/+</sup> and *Taf4b*<sup>−/−</sup> samples to account for updates made to software packages used in our computational pipeline, as well as updated gene annotations in the GO database. We found a slightly lower number of DEGs than we did previously (951 versus 964) but these DEG lists were over 99% identical

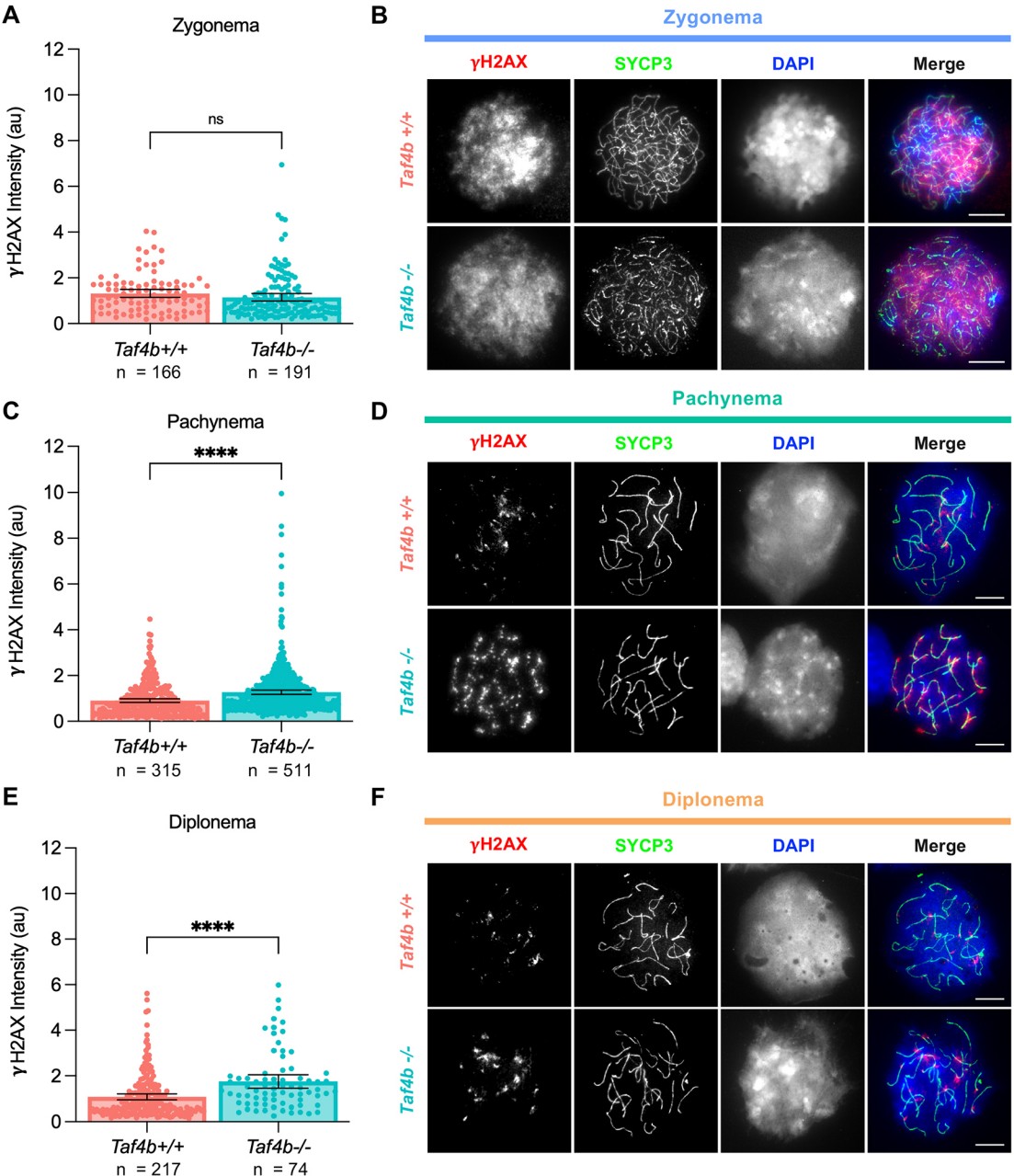

**Fig. 5. *Taf4b*$^{-/-}$ oocytes have elevated levels of γH2AX during pachynema and diplonema.** (A,C,E) Quantification of γH2AX intensity in chromatin spreads during zygonema (A) pachynema (C) and diplonema (E). Spreads were pooled from three E16.5, E18.5 and P0 mice per genotype. *n*, the number of spreads analyzed; au, arbitrary units. (B,D,F) Images of spreads stained with γH2AX (red), SYCP3 (green) and DAPI (blue) from each genotype during zygonema (B), pachynema (D) and diplonema (F). Scale bars: 10 μm. Data are mean±s.e.m. Dots represent individual spread results. The *Taf4b*$^{+/+}$ images in B and F are also shown in Fig 4B. Statistical significance for stage-specific comparisons was determined using an unpaired two-tailed *t*-test. ns, not significant; ****$P$<0.0001.

(Fig. S5A,B, Table S2). Intriguingly, chromatin remodeling was one of the top GO categories present in our E16.5 and E18.5 *Taf4b*$^{-/-}$ DEGs (Fig. 6C and Fig. S5C). Genes present in the chromatin remodeling category differ between E16.5 and E18.5 *Taf4b*$^{-/-}$ DEG lists, but we did find that the H3K4 demethylase *Kdm1b* was consistently downregulated (Fig. S5D). We also found that several histones (*H1f3*, *H1f4*, *H1f6*, *H2ac1*, *H2bc1*, *H2bc22*, *H2bc6*, *H4c8* and *H4c9*) were upregulated at both timepoints but their expression is still very low overall (Fig. S5E). Unfortunately, little is known about the epigenetic landscape of mouse oocytes at E16.5 and E18.5, but these data suggest that changes in chromatin

structure are important for female germ cell development at this stage (Hu et al., 2023). We also found that genes associated with reproductive development, such as *Ddx4*, *Nobox* and *Sohlh1* were consistently downregulated in E16.5 and E18.5 *Taf4b*$^{-/-}$ oocytes (Fig. S5D). Moreover, we found that expression of the X chromosome is also reduced at E18.5, and that there is a significant overlap between *Taf4b*$^{-/-}$ and XO DEGs at this timepoint (Fig. S6A-C). GO analysis of the 642 overlapping DEGs revealed that there was shared disruption of genes involved in meiotic cell cycle, chromatin remodeling and oogenesis (Fig. S6D). Together, these data suggest that *Taf4b*$^{-/-}$ oocytes consistently fail to increase

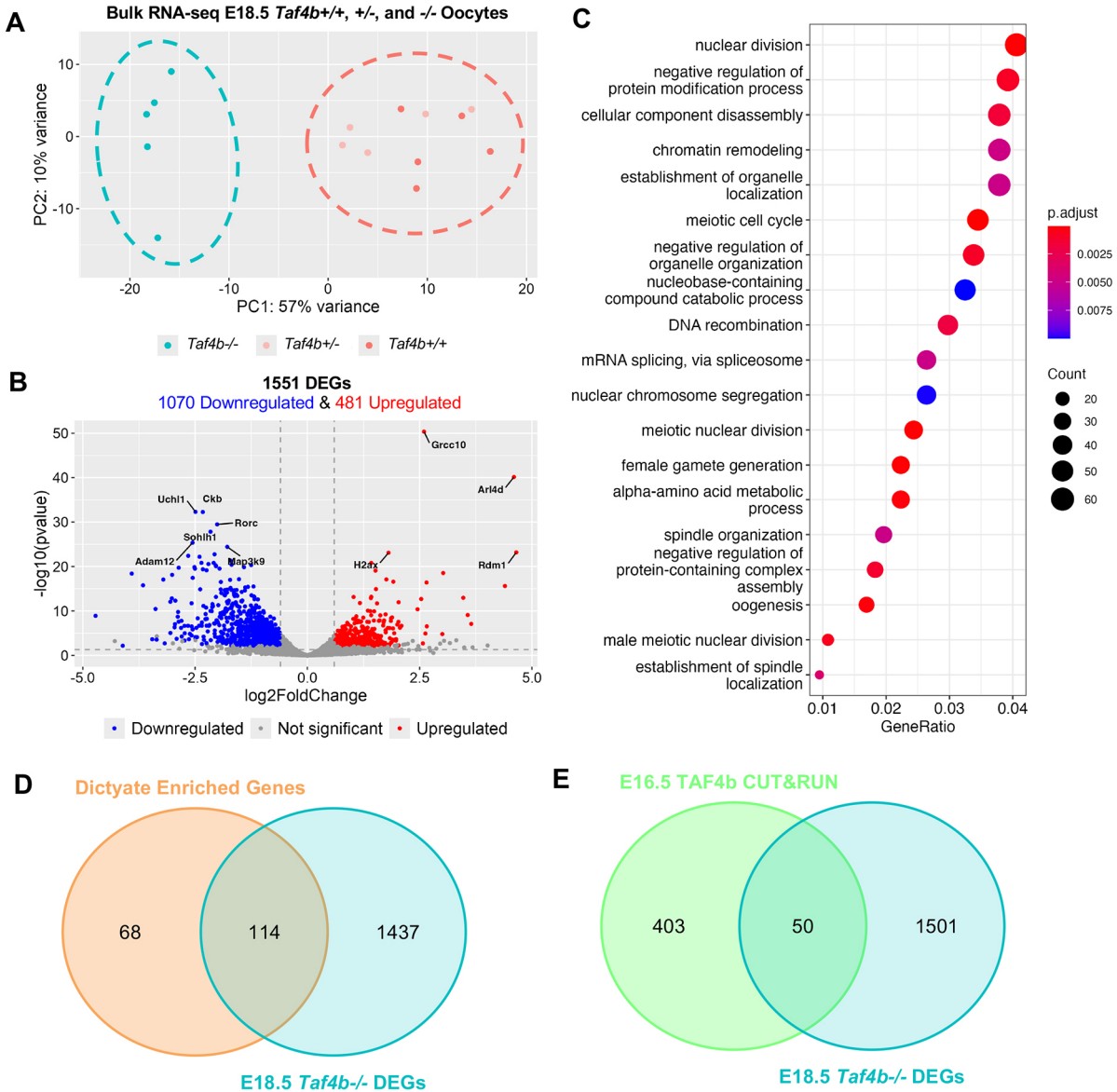

**Fig. 6. Bulk RNA-seq of E18.5 *Taf4b*^+/+, *Taf4b*^+/− and *Taf4b*^−/− oocytes.** (A) PCA plot of E18.5 samples labelled based on genotype. Dashed lines outline samples compared in differential gene expression analysis. (B) Volcano plot of differentially expressed genes (protein-coding, *P*adj<0.05; avg TPM>1; log2FoldChange>l0.6 I). The top 10 most significant DEGs are labelled. Dashed lines represent padj and log2FoldChange cutoffs. (C) Dotplot of GO biological process analysis of all 1551 DEGs. (D) Venn diagram of E18.5 *Taf4b*^−/− DEG list compared with dictyate enriched gene list (Niu and Spradling, 2022). (E) Venn diagram of E16.5 TAF4b CUT&RUN and E18.5 *Taf4b*^−/− DEGs list.

expression of genes required for early oocyte development at both E16.5 and E18.5.

## DISCUSSION

From their origins as mitotic primordial germ cells (PGCs) in the fetal ovary, female germ cells undergo a multitude of precise developmental transitions to achieve meiotic and fertilization competence (Grive and Freiman, 2015). In the early mammalian embryo, bone morphogenetic protein (BMP) signals from extraembryonic mesoderm lead to PGC specification via the expression of three transcription factors: PRDM1, PRDM14 and TFAP2C (Saitou and Yamaji, 2012). After migration to the bipotential gonad, sex specification of the supporting somatic cells into pre-granulosa cells leads to retinoic acid-dependent STRA8 and retinoic acid-independent ZGLP1 expression, which are both germ cell-specific transcriptional regulators of the transition from mitosis to

meiosis (Anderson et al., 2008; Ishiguro, 2023; Miyauchi et al., 2017, 2018; Nagaoka et al., 2020). However, transcription factors that regulate the timing of progression through meiosis I, early oocyte differentiation and survival required for arriving at proper dictyate arrest remain unknown. Moreover, whether these simultaneous but separable genetic processes have common regulators is unknown. Here, using several genomic and developmental approaches, we find that, although TAF4b is not required for meiotic initiation at E14.5, it is essential for the timely progression of meiotic prophase I and its absence largely results in oocytes stalled in pachytene compared to wild-type control oocytes at P0. At the genomic level, TAF4b-deficient pachytene meiocytes display elevated double-strand DNA breaks, fail to turn down early meiotic gene expression and turn up oocyte differentiation gene expression. Delayed and then stalled meiotic prophase I progression likely leads to excessive oocyte attrition at birth and POI-like phenotypes in adult female mice (Fig. 8).

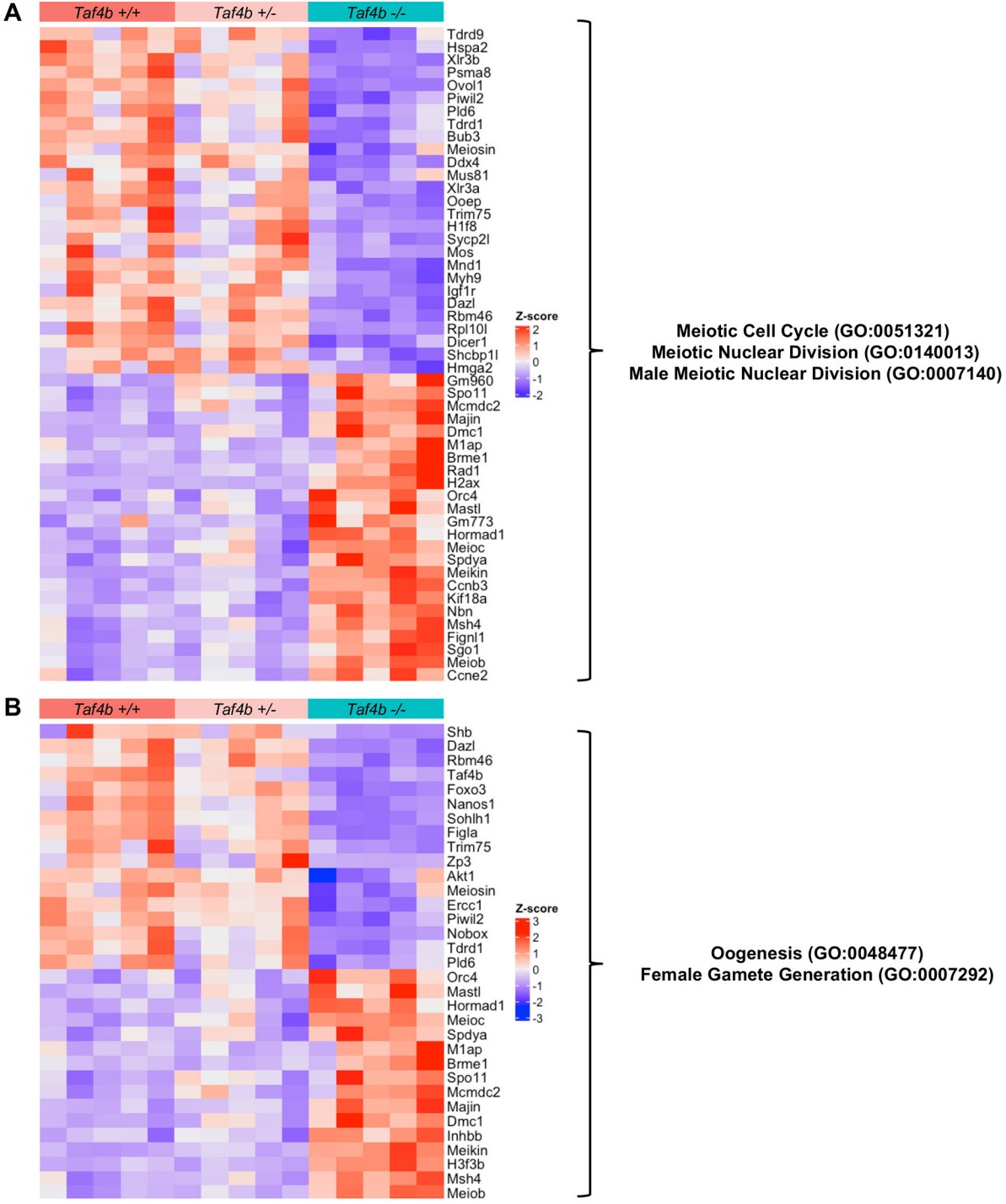

**Fig. 7. Dysregulation of genes involved in meiosis and oogenesis in E18.5 *Taf4b*$^{-/-}$ oocytes.** (A) Heatmap of genes in meiotic cell cycle, meiotic nuclear division or male meiotic nuclear division GO categories that were differentially expressed in E18.5 *Taf4b*$^{-/-}$ oocytes. (B) Heatmap of genes in oogenesis or female gamete generation GO categories that were differentially expressed in E18.5 *Taf4b*$^{-/-}$ oocytes. Heatmaps were generated using based normalized counts output from DESeq2.

Although the fetal human ovary is not easily amenable to such functional studies, there are a number of important reasons to conclude that mouse TAF4b functions are conserved during human oogenesis. First, the dynamic expression of *TAF4b* mRNA in human fetal oocytes is consistent with what we observe in the mouse (Gura et al., 2020). Second, several human genetic studies linked TAF4b expression to POI and human oocyte quality (Di Pietro et al., 2008; Knauff et al., 2009). Recently, a GWAS study of human age at natural menopause demonstrated 290 potential POI-related genes, many of which are mis-expressed in *Taf4b*-deficient mouse E18.5 embryonic oocytes (Ruth et al., 2021). Finally, a heterozygous *TAF4b* mutation is predicted in the genome of the mother of three infertile brothers that display homozygous and truncating mutations in *TAF4b* that cause their non-obstructive azoospermia (Ayhan et al., 2014). Together, our data strongly support that TAF4b plays similar roles during mouse and human gametogenesis.

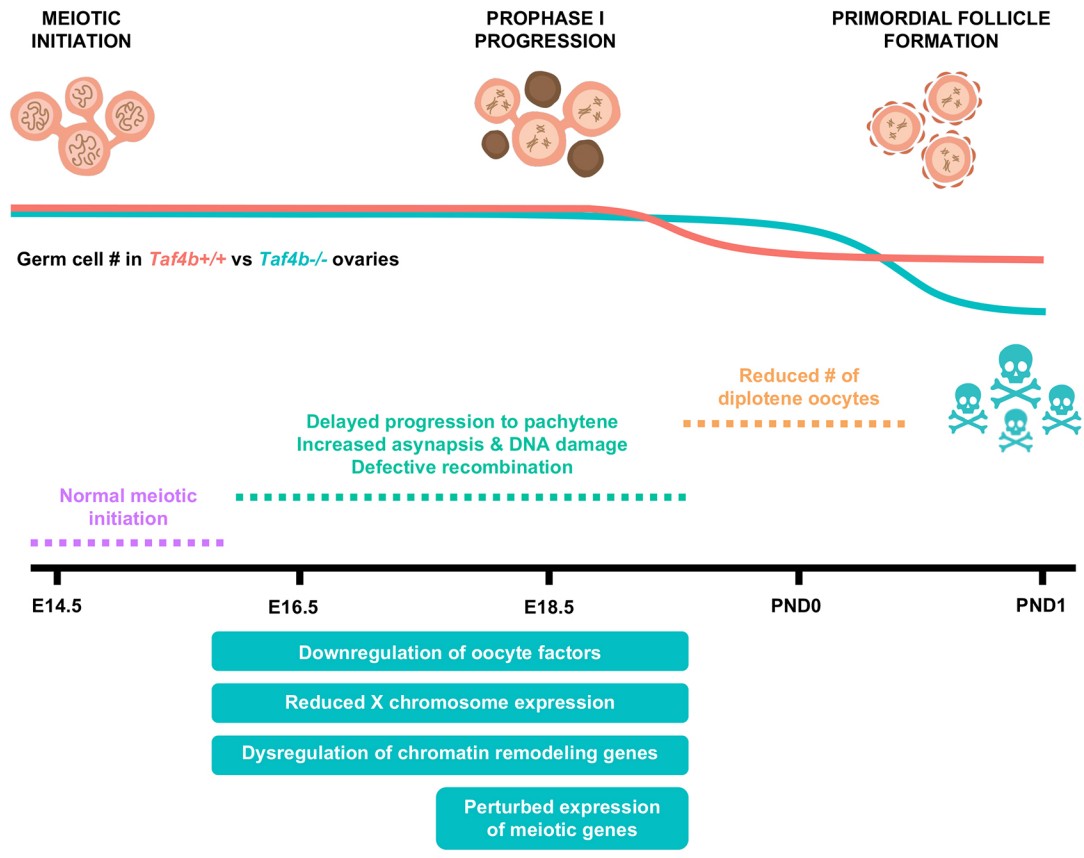

**Fig. 8. Model of defective ovarian reserve establishment in *Taf4b*<sup>−/−</sup> mice.** Schematic timeline of major events that occur during ovarian reserve establishment in mice (top). Estimations of oocyte numbers in *Taf4b*<sup>+/+</sup> (red line) and *Taf4b*<sup>−/−</sup> (blue line) mice throughout the time course are depicted in the schematic below. In the absence of TAF4b, there is a reduced number of oocytes at postnatal day 1. *Taf4b*<sup>−/−</sup> oocytes at E18.5 and postnatal day 0 are unable efficiently to reach diplotene and are largely stuck in pachytene. *Taf4b*<sup>−/−</sup> oocytes do not reach pachytene in a timely manner at E16.5 but initiation of meiosis occurs normally at E14.5. At E16.5 and E18.5, *Taf4b*<sup>−/−</sup> oocytes downregulate genes essential for oocyte development, display reduced expression of the X chromosome and dysregulate chromatin remodeling genes. In addition, expression of genes required for successful completion of prophase is highly perturbed in *Taf4b*<sup>−/−</sup> oocytes at E18.5.

In this study, we have identified that TAF4b functions to promote meiosis I progression through the pachytene-to-diplotene transition in late embryonic mouse oocytes. In addition, we have identified many crucial meiotic and oocyte differentiation genes that are increased and decreased, respectively, in *Taf4b*-deficient E18.5 oocytes compared to controls. We infer from these data that TAF4b both directly and indirectly regulates the expression of these gene programs, which are required for proper homologous recombination, oocyte differentiation and dictyate arrest. However, the precise mechanism by which TAF4b and its associated partners coordinate the integration of all these events is still unknown. Recently, we used CUT&RUN to map its occupancy to the core promoter regions just upstream of the TSS of several hundred E16.5 oocyte genes (Gura et al., 2022). These data are consistent with TAF4b participating in core promoter recognition as part of an embryonic- and germ cell-specific form of TFIID. Our data also reflect the exciting possibility that a TAF4b-containing transcription complexes likely fine-tunes transcription levels (up or down) rather than turning genes on or off. This buffering function is likely important to move oocytes through delicate meiotic transitions with maximum developmental fidelity. Oocyte aneuploidy largely results from meiotic prophase I errors and TAF4b is required for orchestrating the proper timing through pachynema. In the absence of this regulation by TAF4b, meiotic prophase I progression is largely stalled during pachynema and the

majority of such error-prone oocytes are culled from the ovary by excessive oocyte attrition soon after birth.

Finally, several additional levels of gene regulation are known and likely to be coordinated during these early meiotic and oocyte differentiation events. These include chromatin remodeling, and the post-transcriptional and post-translational regulation of RNA and proteins, respectively. TAF4b may help to integrate several diverse aspects of oocyte gene expression. This is most notable in the GO categories revealed to be required during the transition from meiosis to oogenesis (Fig. 6), such as 'protein modification', 'chromatin remodeling' and 'RNA splicing'. In addition, many sequence-specific transcription factors such as *Nobox*, *Figla*, *Sohlh1* and *Foxo3* are dysregulated in the absence of TAF4b and likely have their own effects on gene transcription. Expression of UCHL1, a potent regulator of ubiquitin-mediated proteolytic events in early oocytes, is also reduced (Woodman et al., 2022). Finally, MEIOC, a post-transcriptional regulator of meiotic prophase I progression displays reduced expression in TAF4b-deficient oocytes (Soh et al., 2017). It is likely that these developmental transitions are so delicate and dynamic that it is the concerted integration of gene expression at all these levels, including TAF4b, that ensures that developing oocytes correctly traverse these early meiotic and oocyte differentiation programs in a timely fashion to achieve proper dictyate arrest.

## MATERIALS AND METHODS

### Ethics statement

This study was approved by Brown University IACUC protocol 21-02-0005. The primary method of euthanasia was $CO_2$ inhalation and the secondary method used was cervical dislocation, both as per American Veterinary Medical Association (AVMA) guidelines on euthanasia.

### Mice

Mice that were homozygous for an Oct4-EGFP transgene (The Jackson Laboratory: B6;129S4-Pou5f1$^{tm2Jae}$/J) were backcrossed to the C57BL/6 line and mated for collections used to prepare meiotic spreads. Mice that were homozygous for an Oct4-EGFP transgene (The Jackson Laboratory: B6;129S4-Pou5f1$^{tm2Jae}$/J) and C57BL/6 mice heterozygous for the Taf4b-deficiency mutation (in exon 12 of the 15 total exons of the Taf4b gene that disrupts the endogenous Taf4b gene) were mated for collections used for mRNA isolation, meiotic spread preparation and ovarian immunofluorescence. Timed matings were estimated to begin at day 0.5 by evidence of a copulatory plug. The sex of the embryos was identified by confirming the presence or absence of testicular cords. Genomic DNA from tails was isolated using Qiagen DNeasy Blood & Tissue Kits (69506) for PCR genotyping assays. All animal protocols were reviewed and approved by Brown University Institutional Animal Care and Use Committee, and were performed in accordance with the National Institutes of Health Guide for the Care and Use of Laboratory Animals.

### Immunofluorescence of fetal ovaries

Prenatal ovaries were harvested at E14.5, cleaned of excess fat and fixed in 4% formaldehyde solution for 2 h before embedding in optimal cutting temperature (OCT) compound. Ovaries were serially sectioned at 8 μm on a Thermo-Shandon Cryotome E Cryostat onto positively charged glass slides and washed twice for 5 min in PBST [1× PBS containing 0.1% Tween-20 (ThermoFisher Scientific)] at room temperature. Tissue sections were then incubated in blocking buffer [5% goat serum (Sigma-Aldrich) in PBST] for 1 h at room temperature. Slides were stained with rat anti-GCNA1 (Abcam, ab82527) and rabbit anti-Stra8 (Abcam, ab49602) primary antibodies at 1:100 in blocking buffer and incubated overnight at 4°C on a shaker at low speed. Slides were then washed three times for 10 min each in PBST at room temperature. Slides were stained with goat anti-rat Alexa 488 (Abcam, ab150165) and goat-anti rabbit Alexa 555 (Abcam, ab150086) secondary antibodies at 1:500 in blocking buffer, counterstained with 4′,6- diamidino-2-phenylindole (DAPI; Vector Laboratories) and incubated for 1 h at room temperature. Slides were washed three times for 10 min each in PBST at room temperature before mounting coverslips with Vectashield Antifade Mounting Medium (Vector Laboratories). A secondary antibody only control was included to compare background staining. Images were taken at 20× magnification on a Zeiss Axio Imager M1 microscope.

All images were analyzed using FIJI, and germ cell densities were determined by counting the number of GCNA- or STRA8-positive cells per section and dividing the number by the DAPI-stained area of the ovary to obtain cells/mm$^2$ (Schindelin et al., 2012). The cell counter plugin (Plugins>Analyze>Cell Counter) was used to annotate GCNA- or STRA8-positive cells. The freehand tool was used to outline the tissue section and analyze measurements (Analyze>Measure) was used to determine the area within the outline. For each genotype, three spatially dispersed sections from each ovary obtained from three mice were quantified (3 mice per genotype×2 ovaries per mouse×3 sections per ovary=18 sections per genotype). Results were averaged and significance determined using an unpaired two-tailed t-test. For all graphs, dots represent individual sections counted, bar height represents the sample mean and error bars represent s.e.m. All statistical analyses and graphs were generated using Graphpad/Prism version 10.1.1.

### Preparation of meiotic prophase I spreads

Ovaries were harvested at E16.5, E18.5 or P0 and put in PBS prewarmed at 37°C until their use for spread preparation. Ovaries were then incubated in hypotonic extraction buffer [30 mM Tris, 50 mM sucrose, 17 mM trisodium citrate dihydrate, 5 mM EDTA, 0.5 mM DTT and 0.5 mM phenylmethylsulphonyl fluoride (PMSF; pH 8.2)] for 30 min at 37°C. After HEB incubation, ovaries were gently teased apart with 30-gauge hypodermic needles in 100 mM sucrose at room temperature. The single cell suspension was then pipetted onto slides wet with 1% PFA with 0.2% Triton X-100 and allowed to settle overnight in a humid chamber at 37°C. The next day, slides were air-dried at room temperature for approximately 2-3 h. Slides were washed twice for 5 min each in 0.4% Photo-Flo (Kodak) in PBS and then once for 5 min in 0.4% Photo-Flo in sterile water. After washes, slides were air-dried again for ∼1 h and then stored at −80°C.

### Immunofluorescence of meiotic prophase I spreads

After storage at −80°C, spreads were brought to room temperature by washing in PBST [1×PBS containing 0.1% Tween-20 (Fisher Scientific)] for 15 min. Slides were then incubated in blocking buffer [5% goat serum (Sigma-Aldrich) in PBST] for 45 min at room temperature. Slides were stained with primary antibodies diluted in blocking buffer and incubated overnight at room temperature. Primary antibodies used were mouse anti-SYCP3 (Santa Cruz, sc74659, 1:100 dilution), rabbit anti-SYCP3 (Abcam, ab15093, 1:100 dilution), mouse anti-γH2AX (Millipore, 05-636. 1:100 dilution), rabbit anti-RAD51 (Abcam, ab176458, 1:250 dilution) and rabbit anti-TAF4b (as previously described, Grive et al., 2016, 1:100 dilution). Slides were then washed three times for 5 min each in PBST at room temperature. Slides were stained with secondary antibodies at a 1:500 dilution in blocking buffer for 1 h at room temperature and counterstained with 4′,6- diamidino-2-phenylindole (DAPI; Vector Laboratories). Secondary antibodies used were goat anti-rabbit Alexa 488 (Abcam, ab150081), goat anti-mouse Alexa 488 (Abcam, ab150077), goat anti-rabbit 555 (Abcam, ab150086) or goat anti-mouse 555 (Abcam, ab150118). Slides were washed three times for 5 min each in PBST at room temperature before mounting on coverslips with Vectashield Antifade Mounting Medium (Vector Laboratories). A secondary antibody-only control was included to compare background staining.

Images were taken at 100× magnification on a Zeiss Axio Imager M1 microscope and all image analyses were performed using FIJI. Images of 50-100 spreads were isolated from each slide, and each image was cropped to a 30-50 μm$^2$ square to ensure the area contained only an individual chromatin spread. All analyses were performed on images in .czi file format. Representative images were saved in .tiff format to improve accessibility while maintaining raw data integrity. All image montages shown were generated by first converting the .czi image to RGB (Image>Type>RGB Color) and then converting the RGB to a montage using the RGB to Montage plugin with a greyscale colorization and 10 μm scale bar.

For data shown in Fig. 4, SYCP3 configuration was used to determine prophase I substage. For each timepoint, the total number of spreads analyzed from three animals per genotype were pooled to compare the percentage of spreads in each stage between Taf4b$^{+/+}$ and Taf4b$^{−/−}$ genotypes. A Chi-squared test was performed in Graphpad/Prism (v. 10.1.1) to determine statistical significance of the difference in prophase I substage distribution between genotypes.

A previously published FIJI macro script was used for quantification of fluorescence intensity shown in Figs 2 and 5 (Alexander et al., 2023). Images used for input into the macro were in .czi format with DAPI in blue, SYCP3 in green, and either TAF4b or γH2AX in red. Intensity of DAPI and either TAF4b or γH2AX was measured using Otsu's thresholding. Intensity of TAF4b or γH2AX was normalized to DAPI to account for variations in DNA content among spreads. A copy of the FIJI macro used can be found in Alexander et al. (2023). Fluorescence intensity quantification was performed on all images, and then spreads were grouped based on prophase I substage. An ordinary ANOVA with multiple comparisons was used to compare TAF4b intensity among substage groups (Fig. 2A). An unpaired two-tailed t-test was used to compare intensity between substage groups (Fig. 2B) and genotypes at each substage (Fig. 5). For all graphs, dots represent individual spreads measured, bar height represents the sample mean and error bars represent s.e.m. All statistical analyses and graphs were generated using Graphpad/Prism version 10.1.1.

For data shown in Fig. S3, RAD51 foci were counted manually for each spread. Spreads were then grouped based on prophase I substage, and RAD51 foci counts were compared between Taf4b$^{+/+}$ and Taf4b$^{−/−}$. An unpaired two-tailed t-test was used to compare foci numbers between genotypes at each substage. For all graphs, dots represent individual spreads measured, bar height

represents the sample mean and error bars represent s.e.m. All statistical analyses and graphs were generated using Graphpad/Prism version 10.1.1.

### Embryonic ovary dissociation and fluorescence-activated cell sorting

To dissociate ovarian tissue into a single-cell suspension, embryonic ovaries were harvested, placed in 0.25% Trypsin/EDTA and incubated at 37°C for ~45 min for E18.5, as previously described (Gura et al., 2020, 2022). Eppendorf tubes were subject to vigorous pipetting at 15 min intervals to dissociate tissue throughout incubation. Trypsin was neutralized with FBS. Cells were pelleted at 1500 rpm (221 *g*) for 5 min, the supernatant was removed and cells were resuspended in 100 µl PBS. The cell suspension was strained through a 35 µm mesh cap into a FACS tube (Gibco, 352235). Propidium iodide (ThermoFisher, P3566) was added at a 1:100 dilution to the cell suspension as a live/dead distinguishing stain. Fluorescence-activated cell sorting (FACS) was performed using a Becton Dickinson FACSAria III in the Flow Cytometry and Cell Sorting Core Facility at Brown University. A negative control non-GFP$^+$ mouse tissue was used for each experiment to establish an appropriate GFP signal baseline. Dead cells were discarded and the remaining cells were sorted into GFP$^+$ and GFP$^-$ samples in PBS at 4°C for each embryo.

For RNA-seq analysis, GFP$^+$ cells from each individual embryo were kept in separate tubes and were then spun down at 1500 rpm for 5 min and then resuspended in 150 µl Trizol (ThermoFisher, 1556026). The number of cells for each sample sequenced can be found in Fig. S4. Samples were stored at −80°C.

### RNA-sequencing

Embryonic germ cells resuspended in Trizol were shipped to GENEWIZ/Azenta Life Sciences on dry ice. RNA extraction, sample QC, library preparation, sequencing reactions and initial bioinformatic analysis were conducted at GENEWIZ/Azenta Life Sciences. Total RNA was extracted from cells following the Trizol Reagent User Guide (Thermo Fisher Scientific). Extracted RNA samples were quantified using Qubit 2.0 Fluorometer (Life Technologies) and RNA integrity was checked using Agilent TapeStation 4200 (Agilent Technologies).

A SMARTSeq HT Ultra Low Input Kit was used for full-length cDNA synthesis and amplification (Clontech), and the Illumina Nextera XT library was used for sequencing library preparation. Briefly, cDNA was fragmented, and adaptor was added using Transposase, followed by limited-cycle PCR to enrich and add index to the cDNA fragments. Sequencing libraries were validated using the Agilent TapeStation, and quantified by using a Qubit fluorometer as well as by quantitative PCR (KAPA Biosystems, Wilmington, MA, USA). The sequencing libraries were multiplexed and clustered onto a flowcell on the Illumina NovaSeq instrument according to the manufacturer's instructions. The samples were sequenced using a 2×150 bp paired end (PE) configuration. Image analysis and base calling were conducted by the NovaSeq Control Software (NCS). Raw sequence data (.bcl files) generated from Illumina NovaSeq were converted into fastq files and de-multiplexed using Illumina bcl2fastq 2.20 software. One mis-match was allowed for index sequence identification.

### Bulk RNA-sequencing data analysis

All computational scripts used in this publication are available upon request. RNA sequencing data generated in this study have been deposited in GEO under accession number GSE285194. E16.5 *Taf4b*$^{+/-}$ and *Taf4b*$^{-/-}$ RNA sequencing data have been deposited in GEO under accession number GSE174366.

All raw fastq files were processed on Brown University's high-performance computing cluster. Reads were quality-trimmed and had adapters removed using Trim Galore! (v 0.6.6) with the parameters –nextera -q 10. Samples before and after trimming were analyzed using FastQC (v 0.11.9) for quality and then aligned to the Ensembl GRCm38 using HiSat2 (v 2.2.1) (Pertea et al., 2016). Resulting sam files were converted to bam files using Samtools (v 1.16.1) (Li et al., 2009).

To obtain TPMs for each sample, StringTie (v 2.2.1) was used with the optional parameters -A and -e. A gtf file for each sample was downloaded and, using RStudio (R v 4.3.2), TPMs of all samples were aggregated into one comma separated (csv) file using a custom R script. To create interactive Microsoft Excel files for exploring the TPMs of each dataset, the csv file of aggregated TPMs was saved as an Excel spreadsheet, colored tabs were added to set up different comparisons, and a flexible Excel function was created to adjust to gene name inputs. To explore the Excel files Tables S1 and S2, click on the 'TPM_Quick_Calc' tab highlighted in yellow and type the gene name of interest into the highlighted yellow boxes.

To obtain count tables, featurecounts (Subread v 2.0.2) was used (Liao et al., 2014). Metadata files for the dataset were created manually in Excel and saved as a csv file. These count tables were used to create PCA plots by variance-stabilizing transformation (vst) of the data in DESeq2 (v 1.42.0) and plotting by ggplot2 (v 3.5.0) (Love et al., 2014). DESeq2 was also used for differential gene expression analysis, with count tables and metadata files used as input. We accounted for potential litter effects in our mouse oocytes by setting it as a batch parameter in DESeq2. For the volcano plot, the output of DESeq2 was used and plotted using ggplot2. DEG lists were used for ClusterProfiler (v 4.10.0) input to create dotplots of significantly enriched GO categories for all DEGs, downregulated DEGs and upregulated DEGs. Heatmaps of curated gene lists were generated by using Complex Heatmap (v 2.18.0) using a z-score matrix of normalized counts generated by DESeq2 as input.

### Single cell RNA-sequencing data analysis

Samples from previously deposited datasets GSE136441, GSE128553, GSE130212 and GSE134339 were downloaded from GEO onto Brown University's high-performance computing cluster at the Center for Computation and Visualization. The fastq files from each dataset were aligned using Cell Ranger (v 6.0.0) count and then aggregated using Cell Ranger aggr. The resulting output from aggr was used as input for Seurat (v 3.9.9) in RStudio (R v 4.0.2) (Stuart et al., 2019). Seurat was used to select for high-quality high quality cells based on the following parameters: nFeature_RNA>1000, nCount_RNA<40,000 and percent.mt<20. These data were then passed to Monocle3 (v 0.2.3) for pseudotime analysis, and for generating uniform manifold approximation and projection (UMAP) and gene expression data (Cao et al., 2019; Qiu et al., 2017; Trapnell et al., 2014).

### Acknowledgements

We thank Dr Alison Delong, Dr Ashley Webb, Dr Karen Schindler, Dr Kathryn Grive and Dr Robbert Creton for their helpful input on these studies. We thank the Center for Computation and Visualization (CCV) at Brown University for providing computational resources needed to complete scRNA-seq and RNA-seq data analysis. We are especially grateful to CCV team members Ashok Ragavendran, Joselynn Wallace, Eric Salomaki, August Guang, Paul Cao, Jordan Lawson, Prasad Bandarkar and Prithvi Thakur for their advice and support. We thank Dr Adriana Alexander for their expertise and advice regarding quantitative immunofluorescence of prophase I spreads. We thank Kevin Carlson and the Brown University Flow Cytometry and Sorting Facility for expertise completing the flow sorting of Oct4-EGFP gonads. The Brown University Flow Cytometry and Sorting Facility has received generous support in part by the National Institutes of Health (NCRR Grant No. 1S10RR021051) and the Division of Biology and Medicine, Brown University. As much of our insights were gained by reprocessing publicly available datasets, we greatly appreciate both the researchers that generated and shared the data initially and the respective repositories for making them available.

### Competing interests

The authors declare no competing or financial interests.

### Author contributions

Conceptualization: K.M.A., M.A.G., K.A.S., R.N.F.; Data curation: K.M.A., M.A.B., A.E.K.N., H.E.R., M.A.G., K.A.S., R.N.F.; Formal analysis: K.M.A., A.E.K.N., H.E.R., M.A.G., K.A.S., R.N.F.; Funding acquisition: K.M.A., M.A.G., K.A.S., R.N.F.; Investigation: K.M.A., M.A.B., A.E.K.N., H.E.R., M.A.G., K.A.S.; Methodology: K.M.A., M.A.B., H.E.R., M.A.G.; Project administration: K.A.S., R.N.F.; Resources: K.M.A., R.N.F.; Software: K.M.A., M.A.G., R.N.F.; Supervision: K.A.S., R.N.F.; Validation: K.M.A., R.N.F.; Visualization: K.M.A., R.N.F.; Writing – original draft: K.M.A., R.N.F.; Writing – review & editing: K.M.A., K.A.S., R.N.F.

### Funding

We are grateful to the National Institute of Child Health and Human Development/ National Institutes of Health for their generous support (1F31HD097933 to M.A.G., 1F31HD105340 to K.M.A., and 1R01HD091848 and 1R01HD113567 to R.N.F.). We also thank the United States-Israel Binational Science Foundation (BSF) for their

support to R.N.F. (2019285). Open Access funding provided by Brown University. Deposited in PMC for immediate release.

**Data and resource availability**
E18.5 *Taf4b*$^{+/+}$, *Taf4b*$^{+/−}$ and *Taf4b*$^{−/−}$ RNA sequencing data were generated in this study and have been deposited in GEO under accession number GSE285194. E16.5 *Taf4b*$^{+/−}$ and *Taf4b*$^{−/−}$ RNA sequencing data have been deposited in GEO under accession number GSE174366. All computational scripts regarding bulk RNA sequencing analysis used in this publication are available upon request. All other relevant data and details of resources can be found within the article and its supplementary information.

**Peer review history**
The peer review history is available online at https://journals.biologists.com/dev/lookup/doi/10.1242/dev.205203.reviewer-comments.pdf

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
