## [Peer Review File · Development (Cambridge, England)]

Coordinating meiotic prophase I progression and early oocyte differentiation

Kimberly M. Abt, Myles A. Bartholomew, Anna E. K. Nixon, Hanna E. Richman, Megan A. Gura, Kimberly A. Seymour and Richard N. Freiman
DOI: 10.1242/dev.205203

Editor: Swathi Arur

Review timeline

Original submission:	28 August 2025
Editorial decision:	22 September 2025
First revision received:	3 February 2026
Accepted:	24 February 2026

Original submission

First decision letter

MS ID#: dev.205203

MS TITLE: Coordinating Meiotic Prophase I Progression and Early Oocyte Differentiation

AUTHORS: Kimberly Abt, Myles Bartholomew, Anna Nixon, Hanna Richman, Megan Gura, Kimberly Seymour and Richard Freiman

Dear Dr Freiman,

I have now received all the referees reports on the above manuscript, and have reached a decision. The referees' comments are appended below, or you can access them online: please go to .

The overall evaluation is positive and we would like to publish a revised manuscript in Development, provided that the referees' comments can be satisfactorily addressed. Please attend to all of the reviewers' comments in your revised manuscript and detail them in your point-by-point response. If you do not agree with any of their criticisms or suggestions explain clearly why this is so. If it would be helpful, you are welcome to contact us to discuss your revision in greater detail. Please send us a point-by-point response indicating your plans for addressing the referees' comments, and we will look over this and provide further guidance.

Reviewer 1

Advance summary and potential significance to field

Abt and colleagues demonstrate that after the initiation of meiosis, transcription factor TAF4b supports oocytes' progression through meiotic prophase I and repair of double strand breaks. In late meiotic prophase I, TAF4b leads to the downregulation of meiotic gene expression and upregulation of oocyte differentiation genes. These findings show that TAF4b establishes the size of the ovarian reserve by coordinating meiotic prophase I and oocyte differentiation in the late fetal ovary.

I find the data convincing and conclusions well-supported. However, I believe text edits are needed so that the reader can truly appreciate the conclusion from one section (see Major Point #1). This is

an important study that defines how a transcription factor supports late meiotic prophase I and concurrent cellular differentiation.

Major points:

1. Section starting at line 371: As presented, the conclusion that meiotic initiation occurs normally at E14.5 is based on STRA8+ cell counts (Fig 3A-D). While STRA8 is required for meiotic initiation on a B6 background, its presence is not sufficient to indicate meiotic initiation is proceeding normally. For example, see Ishiguro et al., 2020 PMID: 32032549 and Cheung et al., 2025 PMID: 39817676.

However, the authors previously published E14.5 sorted oocyte RNA-seq data that failed to identify any real differential expression between control and Taf4b knockout oocytes beyond changes in Taf4b expression (Gura et al., 2022). Even a delay in meiotic initiation shows up in the RNA-seq data by E14.5 (Cheung et al., 2025). Yet the E14.5 RNA-seq data are not cited to support the conclusion that meiotic initiation occurs normally.

Based on the published E14.5 RNA-seq analysis (Gura et al., 2022) and the Stra8 counts presented here, I am convinced that meiotic initiation (i.e., entry into meiotic S phase) occurs normally. At a minimum, I suggest the authors cite the E14.5 RNA-seq analysis in this section to remind the reader of these findings. The authors could consider adding a reanalysis of the E14.5 dataset to highlight that meiotic genes are upregulated normally in the absence of Taf4b.

I also suggest the authors define what they mean by meiotic initiation. For some researchers, meiosis initiates prior to meiotic prophase I at meiotic S phase. But for others, meiosis initiates upon entry into meiotic prophase I with the appearance of the leptotene stage. Based on the presented data, it's unclear if the Taf4b knockout oocytes enter the leptotene stage along a normal timeline, but if "meiotic initiation" is defined as entry into meiotic S phase, then this does not impact the validity of the conclusions.

2. For the differentially expressed genes at E18.5, to what extent do they overlap with the TAF4b peaks identified via CUT&RUN at E16.5? What changes could be due to direct TAF4b activity and what changes are independent?

Minor points:

1. Line 133: the Stra8 antibody is listed with an incorrect product number.
2. Lines 345-347: what are these zygotene and pachytene markers?
3. Line 358: Should it be Figure 2A instead of 2A and C?
4. Line 454: Should it be Figure 6B instead of Figure S5B?

Reviewer 2

Advance summary and potential significance to field

The manuscript by Abt et al, "Coordinating Meiotic Prophase I Progression and Early Oocyte Differentiation" investigates the role of transcription factor TAF4b in establishing the ovarian reserve. The authors examine meiotic initiation and progression through meiotic prophase I in Taf4b mutants and find no effect on meiotic initiation but a delay in progression. This work adds to our understanding of how progression through prophase I of meiosis in oocytes is regulated which has not been well studied.

Comments for the author

My concerns, listed below, are relatively minor and revolve around aspects of clarity.

1. Line 47---this paragraph needs some references, currently there are none.
2. Lines 62-65---the authors should elaborate briefly on the Taf4b phenotype, right now it just says it's needed for a healthy ovarian reserve. Are there fewer oocytes or just defective oocytes? If lost, when?

3. Line 79 and 81--should be Dokshin et al 2013.
4. Line 132, 376, 377-----official name of protein is GCNA (not TRA98)
5. Line 398-400 and 568-571---while this shows a delay in progression, can you really say they are arrested at the pachytene stage? Can you look later?
6. Line 402---do they really have more damage? It seems they have the same amount of damage but don't repair it as well.
7. Line 619----I don't think "Figure 1" is meant to be here.
8. Line 624-625—it would be useful to show just the overlay of TAF4b and SYCP3 without DAPI.
9. Line 764---should be indented and with the line above.

Reviewer 3

Advance summary and potential significance to field

In the manuscript by Kimberly Abt et al., the authors reported meiotic defects in oocytes of Taf4b mutant fetal ovaries. Firstly, based on single-cell RNA-seq results combined from previous studies, and the results from antibody staining of meiotic oocytes, it was found that Taf4b are enriched in pachytene fetal germ cells. They further demonstrated that germ cells in Taf4b mutant ovaries could initiate meiosis normally but had defects in meiotic progression. Increased gamma-H2AX intensity was observed in germ cells of mutant ovaries. Through mRNA-seq of germ cells isolated from mutant ovaries, genes and pathways that had a significantly altered expression, in particularly those involved in meiosis regulation and oocyte differentiation, were identified. The manuscript was clearly written; the results were well organized and presented. The following minor concerns need to be addressed by the authors.

In Lines 62-64, the authors stated: "We more recently discovered TAF4b is critical for development of a healthy ovarian reserve in the mouse embryonic ovary". It will be helpful to introduce a little more details of exact phenotypes/defects of follicle formation in Taf4b mutant ovaries. In the present work, the authors showed that germ cells in mutant ovaries had meiosis progression defects, do these germ cells undergo cell death before and during follicle formation as indicated in Figure 8?

References are needed in the second paragraph where cyst breakdown and follicle formation were introduced.

Lines 66-68. What does 'TAF4b-dependent molecular events' refer to?

Lines 86-87. "Here we show that both Taf4b mRNA and protein are enriched during the pachytene stage of meiotic prophase I." please add "in germ cells or fetal oocytes" to enhance clarity.

Figure 1B did not appear in the result section.

Lines 345-347, Figure 1C and D, it was not clear how the authors concluded that "expression of both (Taf4b and Taf7I) increased at the section of the pseudo time course that was similarly enriched with zygotene and pachytene markers". The markers were not defined in this paragraph and graphs.

Figure 2. please label Figure A and B with E16.5, and Figure C and D with E18.5.

Figure 3B, it will be helpful to add antibodies (color codes) in Figure 3B too. In Figure 3C and D and the figure legend, numbers of the ovaries and sections of each genotype used for quantification need to be included. In Line 641 'dots represent individual sections'. Do the authors mean dots present positive oocyte counts on each individual sections?

Line 402, since gammaH2AX primarily represents DNA double strand breaks, it is more accurate to change 'DNA damage' which includes both single and double strand breaks to 'DNA double strand breaks' in this section.

Line 416 and several other places in this section, "We found that Taf4b^{+/+} and Taf4b^{-/-} oocytes had similar levels of gH2AX signal during Zygonema". Please consider replacing 'signal' with 'intensity', since intensity was what measured and presented here.

In Figure 8 (the model), as stated in the figure legend "Estimations of oocyte numbers in Taf4b^{+/+} (red line) and Taf4b^{-/-} (blue line) mice throughout time course are depicted below schematic" . In the model, it appears that Taf4b mutant ovaries have slightly less germ cells compared to wild type ovaries at E14.5; as germ cell number in wild type ovaries decreases around E19.5, germ cells in mutant ovaries remain (i.e. delayed germ cell loss?); germ cells in mutant ovaries undergo rapid cell death around PND0, which results in less primordial follicles in mutant ovaries. Can the authors confirm whether these are the phenotypes observed experimentally. In addition, it has been shown by previous studies that germ cell number decreases continuously in fetal ovaries starting from E14.5 in wild type ovaries. Please consider showing it in the model if the authors observed a similar phenotype in wild type mice too.

First revision

Author response to reviewers' comments

We thank the reviewers for their valuable feedback which has greatly served to strengthen our study. Accordingly, we have comprehensively addressed the points raised and our responses detailed below have been incorporated into the manuscript text. We have included both highlighted and clean versions of the manuscript for ease of review. Line numbers listed in the response to reviewers correspond to the highlighted version of the manuscript.

Reviewer 1

Major point 1: Section starting at line 371: "As presented, the conclusion that meiotic initiation occurs normally at E14.5 is based on STRA8⁺ cell counts (Fig 3A-D). While STRA8 is required for meiotic initiation on a B6 background, its presence is not sufficient to indicate meiotic initiation is proceeding normally. For example, see Ishiguro et al., 2020 PMID: 32032549 and Cheung et al., 2025 PMID: 39817676. However, the authors previously published E14.5 sorted oocyte RNA-seq data that failed to identify any real differential expression between control and Taf4b knockout oocytes beyond changes in Taf4b expression (Gura et al., 2022). Even a delay in meiotic initiation shows up in the RNA-seq data by E14.5 (Cheung et al., 2025). Yet the E14.5 RNA-seq data are not cited to support the conclusion that meiotic initiation occurs normally. Based on the published E14.5 RNA-seq analysis (Gura et al., 2022) and the Stra8 counts presented here, I am convinced that meiotic initiation (i.e., entry into meiotic S phase) occurs normally. At a minimum, I suggest the authors cite the E14.5 RNA-seq analysis in this section to remind the reader of these findings. The authors could consider adding a reanalysis of the E14.5 dataset to highlight that meiotic genes are upregulated normally in the absence of Taf4b. I also suggest the authors define what they mean by meiotic initiation. For some researchers, meiosis initiates prior to meiotic prophase I at meiotic S phase. But for others, meiosis initiates upon entry into meiotic prophase I with the appearance of the leptotene stage. Based on the presented data, it's unclear if the Taf4b knockout oocytes enter the leptotene stage along a normal timeline, but if "meiotic initiation" is defined as entry into meiotic S phase, then this does not impact the validity of the conclusions".

Response: We thank the reviewer for these insightful comments and agree that STRA8 expression alone is not sufficient to demonstrate normal meiotic initiation. To address this concern, we revisited our previously published bulk RNA-seq dataset from FACS-sorted E14.5 oocytes (Gura et al., 2022) and examined normalized expression of key meiotic initiation genes, including *Stra8*, *Rec8*, *Sycp3*, and *Dazl*. As shown in the plots of normalized counts provided for reviewer evaluation, expression of these markers is not significantly altered in *Taf4b*-deficient oocytes at E14.5, consistent with normal entry into the meiotic program. We now cite our prior analysis

directly in the Results section and clarify that “meiotic initiation” is defined here as entry into meiotic S phase rather than leptotene onset. These changes are reflected in extensive text edits to lines 375-391.

Plots below are for review purposes only:

NOTE: Figure provided for reviewer has been removed. It showed reanalysed data from Gura, M. A., Relovska, S., Abt, K. M., Seymour, K. A., Wu, T., Kaya, H., Turner, J. M. A., Fazio, T. G. and Freiman, R. N. (2022) TAF4b transcription networks regulating early oocyte differentiation. *Development*, 149 (3). doi:10.1242/dev.200074.

Major point 2: “For the differentially expressed genes at E18.5, to what extent do they overlap with the TAF4b peaks identified via CUT&RUN at E16.5? What changes could be due to direct TAF4b activity and what changes are independent?”

Response: We thank the reviewer for raising this important point regarding direct versus indirect TAF4b-dependent regulation. To address this, we overlapped TAF4b CUT&RUN peaks identified at E16.5 with E18.5 DEGs observed in *Taf4b*^{-/-} oocytes. This analysis revealed that 50 DEGs that overlap with TAF4b-bound loci, while the majority of DEGs do not, suggesting that both direct and indirect mechanisms contribute to the transcriptional changes observed at E18.5. Notably, several key meiotic genes, including *Meioc*, *Msh5*, and *Mdc1*, show reproducible TAF4b occupancy coincident with H3K4me3 enrichment. We now include this analysis in new figure 6E and supplemental figure 7 in the revised manuscript. We also include a list of all of the 50 overlapping genes in supplementary table 1. These changes are reflected in our text edits made to lines 459-471.

Minor point 1: “Line 133: the *Stra8* antibody is listed with an incorrect product number.”

Response: We thank the reviewer for identifying this, the product number has been corrected.

Minor point 2: “Lines 345-347: what are these zygotene and pachytene markers?”

Response: We used *Msh4* and *Msh5* as well as *Mlh1* and *Mlh3* as zygotene and pachytene markers, respectively. These markers are shown in figure 1C, which is now appropriately referenced in line 349.

Minor point 3: “Line 358: Should it be Figure 2A instead of 2A and C?”

Response: It should be Figure 2A and the text has been corrected.

Minor point 4: “Line 454: Should it be Figure 6B instead of Figure S5B?”

Response: It should say Figure 6B and the text has been corrected.

Reviewer 2

Comment 1: “Line 47---this paragraph needs some references, currently there are none.”

Response: We thank the reviewer for highlighting this oversight and the appropriate references have been added to this paragraph.

Comment 2: “Lines 62-65---the authors should elaborate briefly on the *Taf4b* phenotype, right now it just says it's needed for a healthy ovarian reserve. Are there fewer oocytes or just defective oocytes? If lost, when?”

Response: We have reorganized text in lines 63-70 to address this important point. We mention that there are fewer oocytes present immediately after birth in *Taf4b*-deficient ovaries.

Comment 3: “Line 79 and 81--should be Dokshin et al 2013”.

Response: We thank the reviewer for highlighting this error and the citation has been fixed.

Comment 4: “Line 132, 376, 377-----official name of protein is GCNA (not TRA98)”

Response: We thank the reviewer for highlighting this error and all text mentioning this protein has been fixed.

Comment 5: “Line 398-400 and 568-571---while this shows a delay in progression, can you really say they are arrested at the pachytene stage? Can you look later?”

Response: We thank the reviewer for this thoughtful comment. In PND0 *Taf4b*-deficient ovaries, we observed that a small subset of oocytes progress to diplotene; however, the proportion is markedly reduced compared to controls (Figure 4A). Because most mutant oocytes are lost by PND1 (Grive et al. 2014) it is technically difficult to determine the prophase I substage of oocytes that persist after this point. Given these physiological constraints we have revised the manuscript to use the more conservative term “stalled” rather than “arrest.” These clarifications have been added to the Results and Discussion sections (lines 408 and 593).

Comment 6: “Line 402---do they really have more damage? It seems they have the same amount of damage but don't repair it as well”.

Response: We thank the reviewer for this insightful comment. We agree that our data indicate that the initial formation of meiotic DSBs is normal in *Taf4b*-deficient oocytes; for example, γ H2AX staining at zygonema is comparable between mutants and controls (Figure 5A-B). The elevated γ H2AX staining observed at pachynema and diplonema in *Taf4b*-deficient oocytes therefore reflect persistent unrepaired DSBs, rather than an increase in the number of DSBs initially generated. We have revised the section title and text to clarify that the phenotype represents defective DSB repair and delayed resolution, rather than “excessive DNA damage” (lines 411). This phrasing more accurately reflects the biological mechanism and addresses the reviewer’s concern.

Comment 7: “ Line 619----I don't think "Figure 1" is meant to be here”.

Response: The text has been removed from the legend.

Comment 8: “Line 624-625—it would be useful to show just the overlay of TAF4b and SYCP3 without DAPI”.

Response: We thank the reviewer for this suggestion, and we have removed DAPI from the overlay.

Comment 9: “Line 764---should be indented and with the line above”.

Response 9: We thank the reviewer for bringing this to our attention, and the line indent has been corrected.

Reviewer 3

Comment 1: “In Lines 62-64, the authors stated: "We more recently discovered TAF4b is critical for development of a healthy ovarian reserve in the mouse embryonic ovary". It will be helpful to introduce a little more details of exact phenotypes/defects of follicle formation in Taf4b mutant ovaries. In the present work, the authors showed that germ cells in mutant ovaries had meiosis progression defects, do these germ cells undergo cell death before and during follicle formation as indicated in Figure 8?”

Response: We thank the reviewer highlighting this point, it was also raised by reviewer 2. We have previously shown that *Taf4b*-deficient ovaries have reduced number of oocytes at PND1 and that these oocytes are positive for markers of apoptosis (Grive et al. 2014). These details about the *Taf4b*-deficient phenotype are now mentioned in lines 63-70.

Comment 2: “References are needed in the second paragraph where cyst breakdown and follicle formation were introduced”.

Response: We thank the reviewer for highlighting this oversight and the appropriate references have been added to this paragraph.

Comment 3: “Lines 66-68. What does ‘TAF4b-dependent molecular events’ refer to?”

Response: We thank the reviewer for pointing out the need for clarification. In this context, “TAF4b-dependent molecular events” refers to gene expression programs and regulatory processes that require TAF4b function in oocytes. To address this concern, we changed the text in lines 67-70 to “Furthermore, genes and pathways whose expression depends on TAF4b in oocytes significantly overlap with those disrupted in Turner syndrome and Fragile X-associated POI (FX-POI), two prominent genetic examples of POI (Gura et al., 2022).”.

Comment 4: “Lines 86-87. “Here we show that both Taf4b mRNA and protein are enriched during the pachytene stage of meiotic prophase I.” please add “in germ cells or fetal oocytes” to enhance clarity.”

Response: We agree with the reviewer and the appropriate text has been added to what is now line 90.

Comment 5: “Figure 1B did not appear in the result section”.

Response: We thank the reviewer for bringing this to our attention. We have added an appropriate reference to this figure in the text at lines 336-338 “We found that our pseudotime analysis successfully recapitulated gene expression changes that occur throughout prophase I and that chronological time points align well with pseudotime in the UMAP (Figures 1B-C).”.

Comment 6:” Lines 345-347, Figure 1C and D, it was not clear how the authors concluded that “expression of both (Taf4b and Taf7l) increased at the section of the pseudo time course that was similarly enriched with zygotene and pachytene markers”. The markers were not defined in this paragraph and graphs”.

Response: We thank the reviewer for pointing out the lack of clarity. To clarify, the zygotene and pachytene stages were defined based on expression of canonical meiotic markers along the pseudotime trajectory: *Msh4/5* are expressed during zygonema and early pachynema, while *Mlh1/3* peaks later in pachynema during crossover formation. We have revised the text in lines 348-350 to explicitly define these markers in the paragraph to make it clear that *Taf4b* and *Taf7l* expression coincides with the window enriched for these meiotic markers. We have also added approximate prophase I substages to the color codes in 1C.

Comment 7: “Figure 2. please label Figure A and B with E16.5, and Figure C and D with E18.5”.

Response: We have added the appropriate labels to Figure 2.

Comment 8: “Figure 3B, it will be helpful to add antibodies (color codes) in Figure 3B too. In Figure 3C and D and the figure legend, numbers of the ovaries and sections of each genotype used for quantification need to be included. In Line 641 ‘dots represent individual sections’. Do the authors mean dots present positive oocyte counts on each individual sections?”

Response: We have updated Figure 3B to include the antibodies used along with their corresponding color codes to improve clarity. We apologize for the ambiguity in the figure legend for 3C and D. Each dot represents the positive oocyte count in an individual ovarian section. We have revised the figure legend in lines 664-665 to make this explicit.

Comment 9: “Line 402, since gammaH2AX primarily represents DNA double strand breaks, it is more accurate to change ‘DNA damage’ which includes both single and double strand breaks to ‘DNA double strand breaks’ in this section”.

Response: We thank the reviewer for this suggestion. While we agree that γ H2AX specifically marks double-strand breaks (DSBs), we intentionally retained the broader term “DNA damage” as the section title to provide a clear overview of the phenotype. Within the section, we explicitly describe our analyses using γ H2AX as a marker for unrepaired DSBs (line 422), ensuring the mechanistic details are clear. Thus, we believe that retaining “DNA damage” as the section title is appropriate, while the text accurately specifies the type of lesions analyzed.

*Comment 10: “Line 416 and several other places in this section, “We found that *Taf4b*^{+/+} and *Taf4b*^{-/-} oocytes had similar levels of *g*H2AX signal during Zygonema”. Please consider replacing ‘signal’ with ‘intensity’, since intensity was what measured and presented here”.*

Response: We thank the reviewer for this suggestion. We have changed “signal” to “intensity” in lines 426-428 and line 681.

*Comment 11:” In Figure 8 (the model), as stated in the figure legend “Estimations of oocyte numbers in *Taf4b*^{+/+} (red line) and *Taf4b*^{-/-} (blue line) mice throughout time course are depicted below schematic”. In the model, it appears that *Taf4b* mutant ovaries have slightly less germ cells compared to wild type ovaries at E14.5; as germ cell number in wild type ovaries decreases around E19.5, germ cells in mutant ovaries remain (i.e. delayed germ cell loss?); germ cells in mutant ovaries undergo rapid cell death around PND0, which results in less primordial follicles in mutant ovaries. Can the authors confirm whether these are the phenotypes observed experimentally. In addition, it has been shown by previous studies that germ cell number decreases continuously in fetal ovaries starting from E14.5 in wild type ovaries. Please consider showing it in the model if the authors observed a similar phenotype in wild type mice too.”*

Response: We thank the reviewer for this careful observation. The schematic in Figure 8 is intended as a conceptual model summarizing our experimental findings rather than a precise quantitative plot. Nonetheless, we can clarify the points raised:

1. Germ cell numbers at E14.5: Experimentally, *Taf4b*-deficient ovaries show comparable germ cell numbers to wildtype at E14.5, consistent with normal meiotic onset. We agree that the schematic may give the impression of a slight reduction in mutants at this stage; we have adjusted the figure to reflect similar E14.5 germ cell numbers in wild type and mutant ovaries.
2. Germ cell loss dynamics: In wildtype ovaries, germ cell numbers gradually decrease from E14.5 through PND0, as reported in prior studies. In *Taf4b*-deficient ovaries, germ cell loss is delayed during fetal stages and then occurs rapidly perinatally. We have updated the model to depict gradual germ cell decrease in wildtype ovaries, while maintaining the perinatal spike in mutant cell death, to better reflect experimental observations.
3. Primordial follicle numbers: The lower number of primordial follicles in mutants is accurately captured in the model, consistent with experimental data at PND1 (Grive et al. 2014),

We have revised the figure and legend to clarify that the lines are schematic representations of germ cell dynamics, now explicitly showing gradual fetal germ cell decline in wild type ovaries and the delayed yet rapid perinatal loss in *Taf4b*-deficient ovaries.

Second decision letter

MS ID#: dev.205203R1

MS TITLE: Coordinating Meiotic Prophase I Progression and Early Oocyte Differentiation

AUTHORS: Kimberly Abt, Myles Bartholomew, Anna Nixon, Hanna Richman, Megan Gura, Kimberly Seymour and Richard Freiman

Dear Dr Freiman,

I am happy to tell you that your manuscript has been accepted for publication in Development, pending our standard publication integrity checks.

Reviewer 1

Advance summary and potential significance to field

Abt and colleagues define Taf4b's impact on meiotic prophase I by examining the chromosomal events of meiosis and changes in gene expression. This adds to the field's understanding of how late meiotic prophase I is molecularly regulated.

The authors have sufficiently addressed my prior concerns.

Reviewer 3

Advance summary and potential significance to field

Comments for the author

The authors have addressed the concerns raised in the first round of review. No further concerns from this reviewer.